# LongHalQA: Long-Context Hallucination Evaluation for MultiModal Large Language Models

## Abstract

Hallucination, a phenomenon where multimodal large language models (MLLMs) tend to generate textual responses that are plausible but unaligned with the image, has become one major hurdle in various MLLM-related applications. Several benchmarks have been created to gauge the hallucination levels of MLLMs, by either raising discriminative questions about the existence of objects or introducing LLM evaluators to score the generated text from MLLMs. However, the discriminative data largely involve simple questions that are not aligned with real-world text, while the generative data involve LLM evaluators that are computationally intensive and unstable due to their inherent randomness. We propose LongHalQA, an LLM-free hallucination benchmark that comprises 6K long and complex hallucination text. LongHalQA is featured by GPT4V-generated hallucinatory data that are well aligned with real-world scenarios, including object/image descriptions and multi-round conversations with 14/130 words and 189 words, respectively, on average. It introduces two new tasks, hallucination discrimination and hallucination completion, unifying both discriminative and generative evaluations in a single multiple-choice-question form and leading to more reliable and efficient evaluations without the need for LLM evaluators. Further, we propose an advanced pipeline that greatly facilitates the construction of future hallucination benchmarks with long and complex questions and descriptions. Extensive experiments over multiple recent MLLMs reveal various new challenges when they are handling hallucinations with long and complex textual data.

## 1 Introduction

Multi-modal Large Language Models (MLLMs) (Dai et al., 2024; Hu et al., 2024; **?**; Bai et al., 2023; Liu et al., 2023b; 2024a; Zhu et al., 2023) have achieved great progress in understanding multi-modal contents, by generating detailed descriptions of images, conducting sophisticated, consecutive conversations with humans, etc. Despite the remarkable advancements, MLLMs often experience severe hallucination problems (Yin et al., 2023; Leng et al., 2023; Huang et al., 2023; Zhu et al., 2024; Yue et al., 2024; Bai et al., 2024) by generating textual responses that are not aligned with the corresponding image contents. While hallucinations significantly compromise MLLMs' reliability and applicability in various vision-language tasks and applications, effective and efficient measurement of the hallucination level of MLLMs has become a prerequisite for diagnosis and mitigation of hallucination in MLLMs.

Several related benchmarks (Yifan et al., 2023; Qiu et al., 2024; Jiang et al., 2024; Liu et al., 2024b; Lovenia et al., 2023; Wang et al., 2023; 2024a) have been proposed to gauge the hallucination level of MLLMs in two representative approaches. The first approach conducts discriminative evaluations, where MLLMs are queried with simple questions about whether some objects exist in the image, as illustrated in the upper part of Fig. 1. The second approach conducts generative evaluations, which first apply MLLMs to describe the image and then adopt LLM evaluators to examine whether MLLMs generate hallucinatory content. However, most existing benchmarks share several constraints: 1) Most discriminative benchmarks merely require a yes-or-no answer, which is often too simple to tell much on the cause of hallucinations. 2) Discriminative benchmarks usually come with very short questions like "Is there an {object} in the image?" which are oversimplified and insufficient for

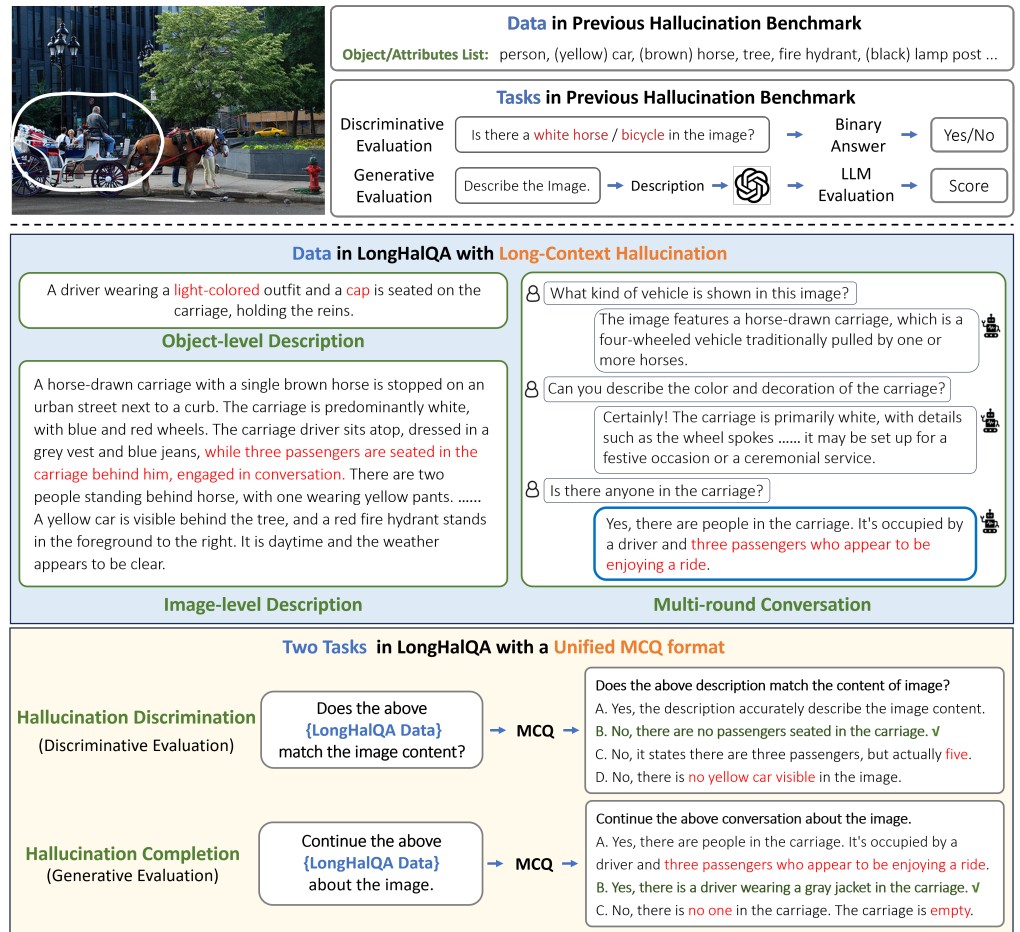

Figure 1: LongHalQA is featured with two novel tasks, namely, *Hallucination Discrimination* and *Hallucination Completion*, which unify both discriminative and generative evaluations into the same multiple-choice-question form without requiring costly LLM evaluations. It comprises three types of long-context data, including *Object-level Description*, *Image-level Description*, and *Multi-round Conversation*. Compared with short and simple questions in existing benchmarks like "Is there an {object} in the image?", the three types of data are more open-ended, richer in contextual information, and closer to real-world data. White circle in image emphasizes the hallucination of passengers.

examining hallucination in sophisticated real-world scenarios. 3) Both discriminative and generative benchmarks (Yifan et al., 2023; Qiu et al., 2024; Kaul et al., 2024) usually leverage off-the-shelf object annotations to construct questions or detect hallucinatory objects, leading to limited variability (e.g., a fixed set of 80 object categories for COCO) and biased evaluations toward a small set of objects. 4) Generative benchmarks (Kaul et al., 2024; Jiang et al., 2024; Qiu et al., 2024; Sun et al., 2023; Liu et al., 2023a) generally employ LLMs for hallucination evaluations, but LLMs are computationally intensive and often unstable due to their inherent randomness.

We design LongHalQA, an LLM-free hallucination benchmark that comprises 6K long and context-rich hallucination questions. LongHalQA is built from GPT4V-generated hallucinatory data that is well aligned with various real-world scenarios. It features two multiple-choice-question (MCQ) tasks, namely, hallucination discrimination and hallucination completion as illustrated at the bottom of Fig. 1. Specifically, hallucination discrimination requires MLLMs to determine whether the given text contains hallucinations and pick the *right causes* of the hallucinations. Hallucination completion instead transforms generative evaluations into a discriminative task, asking MLLMs to continue the text and pick the right option that does not contain hallucinations. LongHalQA thus unifies discriminative and generative evaluations into the same MCQ form, assessing MLLMs' understanding of hallucinations and their tendency to generate hallucinations concurrently. LongHalQA queries come in three data formats as illustrated in the middle of Fig. 1, including object-level descriptions,

image-level descriptions, and multi-round conversation, which are much longer and complex, covering a wide range of 12 types of hallucinations. Such long and complex questions allow LongHalQA to gauge the hallucination levels of MLLMs in more practical applications and scenarios. Compared to existing generative benchmarks, we demonstrate that our MCQ hallucination completion task exhibits similar trends to free-form generative evaluation. Additionally, LonghalQA achieves much higher speed in evaluating MLLMs generative hallucination, especially for extremely large models, facilitating the expansion of evaluation samples and the need for fast testing and evolution of MLLMs.

Additionally, we propose LongHallGen, an automated pipeline for **Long**-context **Hall**ucination Data **Gen**eration. LongHallGen is featured with a set of prompt templates for GPT4v that allow generating hallucination data and converting the generated data into multiple-choice-questions automatically. By modifying the prompt sets, LongHallGen can adjust the type of generated hallucinations, content topics, and data formats. We believe that LongHallGen will serve as a strong basis while creating new or expanding existing hallucination datasets for evaluating and training MLLMs in future research.

Based on LongHalQA, we evaluate ten mainstream MLLMs on long-context hallucinations and provide a comprehensive analysis. The evaluations reveal constraints of MLLMs in discerning and explaining hallucination in long texts, as well as in generating hallucinatory content when completing long texts. Additionally, we observe that the Chain-Of-Thought (COT), a simple but effective hallucination mitigation method (Jiang et al., 2024; Qian et al., 2024), is effective for short queries and generative hallucinations but degrades the performance of most MLLMs on long-context hallucinations discrimination in LongHalQA, especially for those with small sizes, This suggests that COT may be limited by MLLMs' capability on long context processing. We believe that LongHalQA will serve as a basis for mitigating long-context hallucinations in various real-world MLLM tasks.

## 2 RELATED WORKS

**Hallucination Benchmarks for MLLMs.** Various benchmarks have been proposed to measure the hallucination level of MLLMs, including both discriminative and generative benchmarks. For discriminative benchmarks, POPE (Yifan et al., 2023), CIEM (Hu et al., 2023), AMBER (Wang et al., 2023), NOPE (Lovenia et al., 2023), and MME (Fu et al., 2023) query MLLMs with simple questions about the existence or attributes of specific objects in images. PhD (Liu et al., 2024b) and Hal-Eval (Jiang et al., 2024) introduce more types of intrinsic hallucinations into evaluations, such as multi-modal conflicting or event hallucinations. Most of these benchmarks feature simple and short questions and seek solely simple binary "yes" or "no" answers. Generative evaluations introduce LLM evaluators to analyze hallucinations in MLLM-generated text. Various generative benchmarks (Jiang et al., 2024; Kaul et al., 2024; Qiu et al., 2024; Wang et al., 2023) are proposed to improve the scope of evaluated hallucinations and the efficiency and accuracy of LLM evaluators. Our LonghalQA takes a different perspective on long-context hallucinations and encompasses both discriminative and generative evaluations in a unified and efficient MCQ format.

## 3 LONGHALQA: LONG-CONTEXT HALLUCINATION BENCHMARK

LongHalQA comprises 6485 multiple-choice questions (MCQ) that cover two tasks with long-context text: hallucination discrimination and hallucination completion. This section presents the task format in Sec. 3.1, the data format and distributions in Sec. 3.2, and the evaluation metrics in Sec. 3.3.

### 3.1 TASK FORMAT

In order to comprehensively evaluate the hallucination level of MLLMs, we introduce two tasks, namely, hallucination discrimination and hallucination completion, which conduct discriminative and generative MLLM evaluations, respectively.

**Hallucination Discrimination.** For discriminative evaluations, we propose a set of multiple-choice questions to query MLLMs whether object/image descriptions or text responses in a conversation match the contents of images as illustrated in Fig. 1. Each hallucination question is equipped with multiple answer choices and corresponding explanations. One of the choices starts with "yes," suggesting that the text matches the image contents, while the other three start with "no," followed by explanations. MLLMs are required not only to identify the presence of hallucination but also to

Table 1: Statistics of 12 types of hallucinations in LongHalQA. "Object", "Description", and "Conversation" denote the data formats of object-level descriptions, image-level descriptions, and multi-round conversations, respectively. We use both hallucinatory and non-hallucinatory data to construct the discrimination task, while only hallucinatory data for the completion task.

| Hallucination Types | | Object | Description | Conversation | Total |
|---|---|---|---|---|---|
| H1 | (Non) Existent Objects | 234 | 261 | 323 | 818 |
| H2 | Object Attributes | 89 | 130 | 175 | 394 |
| H3 | Object Color | 122 | 90 | 86 | 298 |
| H4 | Object States | 50 | 67 | 84 | 201 |
| H5 | Number of Objects | 80 | 92 | 134 | 306 |
| H6 | Object Locations | 45 | 76 | 86 | 207 |
| H7 | Object Relationships | 49 | 54 | 49 | 152 |
| H8 | Text / Sign Meaning | 27 | 61 | 91 | 179 |
| H9 | Environment Description | 10 | 13 | 31 | 54 |
| H10 | Background Description | 13 | 14 | 21 | 48 |
| H11 | Time | 2 | 7 | 5 | 14 |
| H12 | Weather | 2 | 5 | 13 | 20 |
| Hallucinatory Data | | 723 (52.7%) | 869 (63.3%) | 1098 (68.5%) | 2690 (61.9%) |
| Non-hallucinatory Data | | 647 (47.2%) | 503 (36.7%) | 506 (31.5%) | 1656 (38.1%) |
| **Data for Hallucination Discrimination** | | 1370 | 1372 | 1604 | 4346 |
| **Data for Hallucination Completion** | | - | 869 | 1270 | 2139 |
| Total | | 1370 | 2241 | 2874 | 6485 |

understand why hallucination happens and choose the correct explanation. LongHalQA comes with 4346 such image-question samples for the hallucination discrimination task.

**Hallucination Completion.** Previous generative benchmarks require MLLMs to generate descriptions for the image and employ LLM evaluators to score descriptions. To obviate the slow generation process and costly LLM evaluations, we transform generative evaluations into MCQ format, as shown in Fig. 1. Specifically, we provide an image and a related incomplete description or conversation and ask MLLMs to continue the text. Four answer choices of possible completing sentences are provided, with one correct choice and three hallucinatory choices. Compared to generative benchmarks based on LLM evaluators, the format of generative MCQ significantly reduces evaluation costs and allows for more detailed annotation and analysis of hallucination data. LongHalQA comes with 2149 samples for the hallucination completion task.

The MCQ hallucination completion task simulates the sampling process in MLLM inference, where MLLMs first generate several potential outputs and then select the most appropriate one free from hallucinations. Furthermore, the hallucination completion task can be adapted into a free-form continuation task, where MLLMs are prompted to freely continue long-context data. Our experiments demonstrate that the MCQ format of the hallucination completion task and the free-form generation format yield similar trends in evaluating generative hallucinations of MLLMs.

## 3.2 DATA FORMAT AND DISTRIBUTION

This section presents the format of long-context hallucination data from two aspects, namely, data formats and types of hallucinations, more details to be elaborated in the ensuing subsections.

**Data Formats.** As shown in Tab. 1, LongHalQA consists of three formats of image-text hallucinatory data, including 1370 *Object-level Description*, 1372 *Image-level Description*, and 1604 *Multi-round Conversation*. Specifically, *Object-level Description* describes a specific object in the image, such as its attributes, states, or relations with other objects. *Image-level Description* covers the main contents and more details of an image, such as objects, background, weather, etc, in one paragraph. For the *Multi-round Conversation*, we simulate a human user who communicates with an assistant, querying the image content. The three types of data formats are highly compatible with the actual application scenarios of MLLMs and thus can better simulate real hallucination situations.

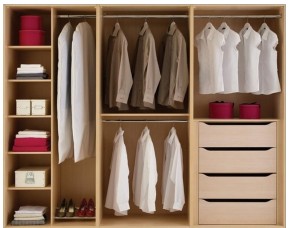

Continue the following conversation:
User: Can you describe what's on the table in this image?
Assistant: The table is set for a meal with blue and yellow patterned plates.

A. There are four such plates in total, and each has a different arrangement of food.
B. There are four such plates in total, and they are all empty.
C. There are five such plates for salads placed around the table.
D. There are five plates in total, and they all have various types of food.

Does the following description match the image content?
"The image displays an organized wardrobe with various compartments containing clothes and accessories. On the left section, there are folded items on shelves including white and crimson towels, ...... The central part shows two levels of hanging space: the upper rack holds beige shirts, while the lower section displays similar items. The right section contains a series of lighter clothes, predominantly white shirts, arranged in two rows, hanging neatly above each other. ......"

Figure 2: LongHalQA involves complex hallucination annotations involving logic and textual consistency, which are closer to hallucinations in real-world MLLM application scenarios.

**Types of Hallucination.** As shown in Tab. 1, We analyze the description texts of objects/images and categorize the wrong descriptions into twelve types of hallucinations. The detailed definitions and visualization of each type of hallucination are provided in Appendix A. Compared to existing benchmarks that focus on the existence and attributes of objects, LongHalQA contains a much broader collection of hallucinations with detailed annotations.

In addition, LongHalQA includes hallucination samples involving logic and contextual consistency, which are untouched in most existing benchmarks. Such complex hallucinations are often observed with contextually inconsistent descriptions such as 'four such plates' or 'five plates' as illustrated in the upper part of Fig. 2, or incorrectly mixed descriptions such as 'two rows of shirts in the central part' vs 'one row of shirts in the right part' as illustrated in the lower part of Fig. 2. We foresee that LongHalQA will inspire more in-depth studies of MLLMs regarding such complex hallucinations.

**Complexity and Length of Text in LongHalQA.** We derive certain statistics of LongHalQA data to verify their quality, including the length and the number of object nouns appearing in the text data. The study shows LongHalQA has an average of 14/130 and 189 words, respectively, for object/image-level descriptions and multi-round conversations, clearly longer and more informative than descriptions of around 80 words in existing generative benchmarks (Jiang et al., 2024; Kaul et al., 2024). In addition, LongHalQA contains approximately 4K object names, indicating more informative data compared with existing benchmarks (Yifan et al., 2023; Hu et al., 2023) with fixed annotations (e.g., 80 object names within COCO dataset).

### 3.3 EVALUATION METHODS AND METRICS

We adopt both binary and multiple-choice settings for the Hallucination Discriminiation task, and multiple-choice setting for Hallucination Completion task. For binary answers, we use Accuracy, Precision, and "Yes" ratios as metrics following previous practices (Yifan et al., 2023; Jiang et al., 2024). For multiple-choice setting, we adopt (mc-)accuracy (Liu et al., 2023d) as the evaluation metric, which requires MLLMs to generate the letter (e.g., A, B, C, or D) of the answer option. We randomly shuffle the order of the four options for each MCQ to reduce the impact of option order.

## 4 LONGHALLGEN: AUTOMATED LONG-CONTEXT HALLUCINATION DATA GENERATION

Given the lack of long-context image-text hallucination data in the broad area of vision language understanding, we dig deep into the proposed LongHallQA and distill LongHallGen, a generic pipeline that aims to facilitate the construction of long-context hallucination benchmarks or datasets in various multimodal tasks, more detailed processes to be elaborated in the ensuing subsections.

Table 2: Evaluations of MLLMs on LongHalQA with Hallucination Discrimination and Hallucination Completion tasks with binary answer accuracy ('bi-Acc') and multi-choice accuracy ('mc-ACC').

| Model | Hallucination Discrimination | | Hallucination Completion | Average |
| | bi-Acc. | mc-ACC. | mc-ACC. | |
|---|---|---|---|---|
| MiniCPM-V2-2B | 44.15 | 40.80 | 46.25 | 43.73 |
| Qwen2-VL-2B | 48.21 | 38.76 | 50.36 | 45.78 |
| Fuyu-8B | 43.31 | 23.86 | 23.67 | 30.28 |
| LLaVA-1.5-7B | 38.52 | 35.04 | 36.08 | 36.55 |
| LLaVA-1.5-13B | 41.83 | 43.60 | 37.58 | 41.00 |
| LLaVA-1.6-7B | 44.13 | 45.56 | 43.40 | 44.36 |
| Qwen-VL-Chat | 43.21 | 37.03 | 36.57 | 38.94 |
| LLaVA-1.6-34B | 46.99 | **57.40** | 56.03 | 53.47 |
| Qwen2-VL-72B | 50.36 | 54.78 | **61.50** | **55.55** |
| GPT4o | **52.80** | 47.63 | 56.15 | 52.19 |

**Image Collection and Filtering.** To generate informative hallucinatory data, the first step involves selecting images that contain rich content and discarding those with overly simple scenes or rare objects. This can be simply achieved by leveraging off-the-shelf image understanding techniques. We adopt images from the validation set of VisualGenome (Krishna et al., 2017) and Objects365 (Shao et al., 2019) to avoid images being used for training MLLMs, and then analyze and filter them based on dataset annotations and GroundingDINO (Liu et al., 2023c). This straightly leads to 1200 images with complex contents for further hallucinatory data generation.

**Positive Data Generation.** With selected images, long-context data can be generated by leveraging state-of-the-art MLLMs. This process involves designing certain prompts according to specific tasks and domains. We adopt GPT4V to generate long-context texts. Unified prompt templates are designed to allow adjusting the format and scope of generated texts, as illustrated in Appendix B. Note current MLLMs, even GPT4V, suffer from severe hallucinations while generating long-context data. A hallucination check process is required to ensure the quality of the generated data.

**Hallucination Check.** The MLLM-generated data are then examined comprehensively for detecting any inherent hallucinations. The examination can be achieved in three steps. First, the generated data undergo a per-sentence self-check by GPT4V, where each piece of data is checked twice to reduce randomness. Second, names of objects present in the data are extracted, and certain image understanding tools such as GroundingDINO (Liu et al., 2023c) are then conducted with detection results feeding to GPT4V for further checking. Finally, the summarized analysis are revised manually, and the revised analysis can serve for the generation of hallucination-explanation data pairs.

Though MLLMs such as GPT-4V can generate lengthy descriptions, they tend to generate numerous hallucinations as well. Take LongHallQA as an example. Among its generated descriptions for the 500 images on image-level descriptions, 394 descriptions (78.8%) contain at least one hallucination. The ratio goes up to 82.4% for the generated conversations. One major cause of the high hallucination rate is due to the increased length of the generated descriptions. In addition, most selected images in LongHalQA have complex scenes, further boosting the possibility of descriptive hallucinations. Nevertheless, such data realistically simulate the hallucinations in actual applications of MLLMs.

**Hallucination-Explanation Pair Generation.** Two different prompts are formulated to construct hallucination-explanation (HE) data pairs with the checking outcome. For the data without hallucinations detected, GPT4V is prompted to modify the data to produce a misleading error within the range of hallucination types suggested in the prompt. For the data containing hallucinations, GPT-4V is prompted to modify the data to contain only one error to form HE pairs. The generated HE pairs are then adopted to construct MCQs for MLLM evaluations.

**Question and Answer Generation.** With the generated HE pairs, MLLMs such as GPT-4V can be employed to generate questions for tasks such as hallucination discrimination and hallucination completion in LongHalQA. For discriminative tasks, questions like 'Does the following {Hallucination Data} match the image?' can be formulated to prompt MLLMs to generate four candidate options with explanations. For completion tasks with questions like 'Complete the following {Hallucination

Table 3: Experiments on LongHalQA for **Hallucination Discrimination** task with binary answers."Acc.", "Pre.", and "YR" denote accuracy, precision, and Yes ratio, respectively.

| Model | Object-level Description | | | Image-level Description | | | Multi-round Conversation. | | |
|---|---|---|---|---|---|---|---|---|---|
| | Acc. | Pre. | YR | Acc. | Pre. | YR | Acc. | Pre. | YR |
| MiniCPM-V2-2B | 59.71 | 54.67 | 74.23 | 36.66 | 36.64 | 99.85 | 36.10 | 31.91 | 89.28 |
| Qwen2-VL-2B | 64.31 | 58.01 | 71.97 | 36.88 | 36.74 | 99.78 | 43.45 | 32.05 | 69.45 |
| Fuyu-8B | 50.29 | 48.52 | 83.65 | **43.29** | 37.28 | 78.79 | 36.35 | 30.72 | 82.79 |
| LLaVA-1.5-7B | 45.18 | 45.99 | 94.60 | 36.59 | 36.62 | 99.93 | 33.79 | 47.53 | 94.58 |
| LLaVA-1.5-13B | 52.70 | 49.96 | 84.16 | 36.95 | 36.95 | 99.71 | 35.85 | 31.93 | 90.02 |
| LLaVA-1.6-7B | 60.51 | 55.31 | 72.85 | 37.10 | 36.82 | 99.56 | 34.79 | 31.90 | 92.83 |
| Qwen-VL-Chat | 58.69 | 53.71 | 79.64 | 36.66 | 36.66 | 100.0 | 34.29 | 31.78 | 93.58 |
| LLaVA-1.6-34B | 68.61 | 61.65 | 67.96 | 38.26 | 37.22 | 98.10 | 34.10 | 32.16 | 96.13 |
| Qwen2-VL-72B | 71.60 | 64.46 | 65.11 | 40.08 | 37.87 | 95.84 | 39.40 | 33.59 | 88.34 |
| GPT4o | **73.94** | 68.31 | 57.81 | 37.92 | 37.13 | 97.03 | **46.32** | 35.71 | 77.74 |

Data}.', MLLMs are employed to construct a completion task by providing prefix text from HE pairs and options of candidate sentences for completion.

LongHallGen exploits MLLMs for most processes in generating long-context hallucination data, except the hallucination checking that involves optional human verification. We expect LongHallGen to serve as a basis for constructing more long-context hallucination data for training and evaluating MLLMs, thereby enhancing their capability and reliability in complex application scenarios.

Table 4: Experiments on **Hallucination Discrimination** under multi-choice settings. "Desc" indicates description.

| Accuracy | Image Desc. | Conversation |
|---|---|---|
| MiniCPM-V2-2B | 39.65 | 41.96 |
| Qwen2-VL-2B | 41.55 | 35.97 |
| Fuyu-8b | 23.47 | 24.25 |
| LLaVA-1.5-7B | 37.17 | 32.92 |
| LLaVA-1.5-13B | 45.99 | 41.21 |
| LLaVA-1.6-7B | 49.42 | 41.71 |
| Qwen-VL-Chat | 37.97 | 36.10 |
| LLaVA 1.6-34B | **60.93** | 53.86 |
| Qwen2-VL-72B | 53.57 | **55.98** |
| GPT-4o | 46.57 | 48.69 |

Table 5: Experiments on **Hallucination Completion** under multi-choice settings."Desc" indicates discription.

| Accuracy | Image Desc. | Conversation |
|---|---|---|
| MiniCPM-V2-2B | 44.07 | 48.43 |
| Qwen2-VL-2B | 47.18 | 53.54 |
| Fuyu-8b | 23.25 | 24.09 |
| LLaVA-1.5-7B | 32.80 | 39.37 |
| LLaVA-1.5-13B | 31.53 | 43.62 |
| LLaVA-1.6-7B | 39.47 | 47.32 |
| Qwen-VL-Chat | 33.14 | 40.00 |
| LLaVA-1.6-34B | 53.16 | 58.90 |
| Qwen2-VL-72B | **59.38** | **63.62** |
| GPT-4o | 50.97 | 61.33 |

## 5 EXPERIMENTS

### 5.1 OVERALL EXPERIMENTS

We adopt LMMs-Eval (Li* et al., 2024) to employ LongHalQA to gauge the hallucination level of MLLMs. The evaluations are performed over nine widely adopted open-source MLLMs, including MiniCPM-V2 (Hu et al., 2024), Qwen series (Bai et al., 2023; Wang et al., 2024b), Fuyu (Bavishi et al., 2023), LLaVA series (Liu et al., 2023b; 2024a), and the closed-source GPT-4o, covering MLLMs' sizes from 2B to 72B and larger. We present the overall experiments in Tab. 2. We use GPTQ-Int8 quantization for Qwen2-VL-72B due to memory constraints. Notably, Qwen2-VL-72B achieves the best accuracy on average for hallucination completion tasks, demonstrating its superior ability to identify hallucinated information and produce reliable content. Such performance advantage may be attributed to the superior capabilities of LLM and their proposed naive dynamic resolution mechanism. Following Qwen2-VL-72B are LLaVA-v1.6-34B and GPT-4o. This performance advantage suggests GPT's potential capability of self-correction for hallucinations, given that the LongHalQA is primarily based on hallucination data from GPT. Next, smaller models like MiniCPM-V2, Qwen2-VL-2B, and

and LLaVA 1.6-7B also achieved excellent results, surpassing many larger models. It is worth noting that both MiniCPM-V2 and Qwen2-VL-2B adopts reinforcement learning to mitigate hallucinations, indicating that this is an effective method for improving the reliability of MLLMs. One common feature of these leading MLLMs is that they all support high-resolution images, suggesting that resolution plays a significant role in alleviating hallucinations.

## 5.2 EXPERIMENTS ON HALLUCINATION DISCRIMINATION

**Binary-Answer Setting.** Tab. 3 shows experiments on the hallucination discrimination task under the binary answer setting. GPT-4o performs the best among all MLLMs, particularly for the multi-round conversations, with an accuracy gain of 9.5% over other MLLMs. Fuyu-8b shows superior capabilities in identifying hallucinations in long text and achieves the best accuracy among all open-source MLLMs, scoring 45.0% for image-level descriptions and 36.8% for multi-round conversations. We observe that most MLLMs produce a high yes ratio of over 70%, even 99% for image-level description, largely deviating from the ratio of non-hallucinatory data with answers 'yes' in LongHalQA (38.1%). Moreover, Qwen-VL-Chat, MiniCPM-V2-2B, and LLaVA series exhibit unbalanced capabilities in handling text of varying lengths. For object-level descriptions, LLaVA1.6-7B achieves an accuracy of 60.6%, but this drops to 37.1% and 34.7% for image-level descriptions and conversations that are about ten times longer. The lower accuracy, coupled with a significantly high Yes ratio, demonstrates the constraints of existing MLLMs in recognizing hallucinations in long contexts.

Table 6: Experiments on different types of hallucinations on discrimination task with multiple-choice setting. The indexes of hallucination types are consistent with Tab. 1.

|  | H1 | H2 | H3 | H4 | H5 | H6 | H7 | H8 | H9 | H10 | H11 | H12 |
|---|---|---|---|---|---|---|---|---|---|---|---|---|
| MiniCPM-V2-2B | 20.1 | 19.5 | 16.5 | 18.3 | 22.7 | 20.7 | 21.8 | 16.7 | 16.7 | 16.7 | 33.3 | 17.6 |
| Qwen-VL-Chat | 12.4 | 10.1 | 10.0 | 9.8 | 10.0 | 12.9 | 11.9 | 8.3 | 16.7 | 6.7 | 33.3 | 11.7 |
| Fuyu-8B | 27.9 | 26.9 | 30.1 | 25.3 | 26.3 | 22.0 | 20.8 | 27.8 | 26.2 | 30.0 | 40.0 | 29.4 |
| LLaVA-1.5-7B | 9.0 | 9.1 | 8.8 | 4.2 | 10.0 | 8.8 | 12.9 | 6.9 | 19.0 | 10.0 | 20.0 | 5.9 |
| LLaVA-1.5-13B | 22.0 | 22.9 | 26.6 | 23.2 | 26.4 | 30.8 | 26.7 | 26.4 | 35.7 | 26.7 | 26.7 | 35.3 |
| LLaVA-1.6-7B | 32.8 | 40.4 | 40.2 | 30.9 | 30.8 | 43.4 | 43.5 | 38.8 | 47.6 | 20.0 | 20.0 | 47.0 |

**Multiple-choice Setting.** Tab. 4 shows experiments on the hallucination discrimination task under the MCQ setting. LLaVA-1.6-34B and Qwen2-VL-72B achieve the highest accuracy in description and conversation data formats, respectively, followed closely by GPT-4o. Notably, most MLLMs achieve much higher accuracy in both image description and conversation formats compared to the accuracy under binary settings. This is likely because answer choices include detailed explanations for the involved hallucinations, giving the model a clearer understanding and aiding in selecting the correct option. However, the much lower ranking-based accuracy suggests that MLLMs struggle to correctly discern hallucinations and provide accurate reasons when they cannot directly access all options for reference, consistent with the low accuracy observed in binary settings where explanations are also inaccessible. Interestingly, Fuyu-8B achieves the highest ranking-based accuracy, even surpassing its generation-based accuracy, which may be attributed to its unique decoder-only structure.

## 5.3 EXPERIMENTS ON HALLUCINATION COMPLETION

Tab. 5 shows experiments on the hallucination completion task. Among open-sourced MLLMs, Qwen2-VL-72B achieves the best performance for both image description and multi-round conversation. Two small models, MiniCPM-V2 and Qwen2-VL-2B, also achieved excellent results in this task, ranking just behind three much larger models with the least size of 34B. This further reflects the significance of high-resolution representation and multi-modal RLHF (Yu et al., 2023), which aligns vision and language for trustworthy behavior against object hallucinations in training. Other MLLMs achieve similar ranking performance as those in the hallucination discrimination task, with LLaVA 1.6-7B prevailing, followed by Qwen-VL-Chat and LLaVA 1.5-13B.

Table 7: Experiments on LongHalQA with the modified prompt. 'bi-ACC' and 'mc-ACC' denote binary answer and multiple-choice accuracies. "Object" and "Long" indicate data formats of object-level descriptions and long context data of image-level descriptions and conversations.

| Model | Hall. Discrimination | | | Hall. Completion | Average |
| | (Object) bi-Acc. | (Long) bi-Acc. | mc-ACC. | mc-ACC | |
| --- | --- | --- | --- | --- | --- |
| MiniCPM-V2 | 63.87(+4.14) | 36.24 (-0.14) | 38.12 (-2.68) | 48.13(+1.88) | 43.90(+0.17) |
| Qwen2-VL-2B | 61.61 (-2.70) | 38.06 (-2.10) | 34.95 (-3.81) | 50.75(+0.39) | 43.87(-1.91) |
| Fuyu-8B | 47.23 (-3.06) | 34.16 (-5.66) | 23.74 (-0.12) | 23.35 (-0.32) | 28.55 (-1.73) |
| LLaVA-1.5-7B | 49.64 (+4.46) | 34.84 (-0.35) | 34.67 (-0.37) | 38.81 (+2.73) | 37.75 (+1.2) |
| LLaVA-1.5-13B | 54.38(+1.68) | 36.29 (-0.11) | 44.12(+0.52) | 40.90(+3.32) | 42.45(+1.45) |
| LLaVA-1.6-7B | 60.36 (-0.15) | 34.68 (-1.26) | 40.68 (-4.88) | 44.48(+1.08) | 42.80 (-1.56) |
| Qwen-VL-Chat | 59.85(+1.16) | 34.76 (-0.71) | 35.38 (-1.65) | 39.06(+2.49) | 39.19(+0.25) |
| LLaVA-1.6-34B | 66.64 (+1.97) | 36.03 (-0.15) | 45.56 (-11.84) | 57.49(+1.46) | 49.76 (-3.71) |
| Qwen2-VL-72B | 73.58(+1.98) | 46.08(+6.34) | 53.11 (-1.67) | 64.82(+3.32) | 57.73(+2.18) |
| GPT-4o | 74.84(+0.90) | 45.50 (+3.38) | 50.12(+2.49) | 58.97(+2.82) | 54.79(+2.6) |

## 5.4 ANALYSIS OF HALLUCINATION TYPES.

We conduct a detailed analysis of different types of hallucinations in the discrimination task as shown in Tab. 6. We find that most MLLMs exhibit relatively higher accuracy in hallucinations of object existence (H1), attributes (H2), and colors (H3), which are relatively simple to discern because they can be directly observed from images and rely less on detailed comprehension of image content. From the perspective of MLLMs, MiniCPM-V2-2B and Qwen-VL-Chat show balanced strength across different types of hallucinations. Fuyu-8B is competitive across multiple types, but struggles with object location (H6) and relationships (H7). LLaVA1.6-7B outperforms other MLLMs on most types of hallucination, especially for object location (H6) and Text/Sign semantic meanings (H8).

## 5.5 PROMPT ANALYSIS

We additionally examine the Chain-Of-Thought (COT) on LongHalQA, which has been verified in previous benchmarks (Qian et al., 2024; Jiang et al., 2024) for mitigating hallucinations. We modify the prompt to guide MLLMs to think step by step, and provide more instructions, such as possible types of hallucinations and suggestions for per-sentence verification (Refer to Appendix C for details). As shown in Tab. 7, apart from GPT-4o, most MLLMs experience a drop in performance across different tasks when using Chain of Thought (COT)—notably, the larger the language model, the smaller the performance drop. Qwen2-VL-72B only experienced a drop in the multiple-choice task for hallucination Discrimination, with an average increase of 2.18 accuracy. We conjecture that this is largely due to the limited capability of MLLMs to interpret long-context data. Besides, the modified prompts improve short query discrimination and hallucination completion the most but have little effect on discriminate hallucinations in long texts.

## 6 COMPARISON WITH FREE-GENERATION EVALUATION

We further compare LongHalQA with evaluations in free-generation scenarios to examine whether the multiple-choice (MCQ) format of the hallucination completion task in LongHalOA accurately captures the true generative capabilities of MLLMs. As described in Sec. 3.1, the multiple-choice hallucination completion task could be transformed into a similar counterpart in a free-form generation setting. Specifically, we randomly selected 200 image-text data from image description and multi-round conversation data for the hallucination completion task, respectively, and apply MLLMs to complete the text freely. As shown in Tab. 8, the ranking of hallucination levels, using GPT-4 evaluation, is largely consistent with the results from LongHalQA, demonstrating that the proposed MCQ task is able to capture the generative capabilities of MLLMs.

Additionally, we compare the efficiency of different approaches in evaluating MLLM's generative hallucination. As shown in Fig fig. 3, for some extremely large models, the time required for

Table 8: Comparison of Multi-Choice (mc-ACC) and Free-Generation (gen-ACC) settings for **Hallucination Completion**. Under the free-generation setting, MLLMs are provided with preceding contexts and images and are prompted to freely continue the pretext to assess their generative hallucinations.

| Accuracy | mc-Acc. | Ranking | gen-Acc. | Ranking |
|---|---|---|---|---|
| MiniCPM-V2 | 46.25 | 4 | 54.00 | 5 |
| Qwen2-VL-2B | 50.36 | 3 | 55.25 | 4 |
| Fuyu-8b | 23.67 | 9 | 11.50 | 9 |
| LLaVA 1.5-7B | 36.08 | 8 | 52.50 | 7 |
| LLaVA 1.6-7B | 43.40 | 5 | 59.50 | 3 |
| LLaVA 1.5-13B | 37.58 | 6 | 53.50 | 6 |
| Qwen-VL-Chat | 36.57 | 7 | 50.50 | 8 |
| LLaVA 1.6-34B | 56.03 | 2 | 63.00 | 2 |
| Qwen2-VL-72B | 61.50 | 1 | 65.75 | 1 |

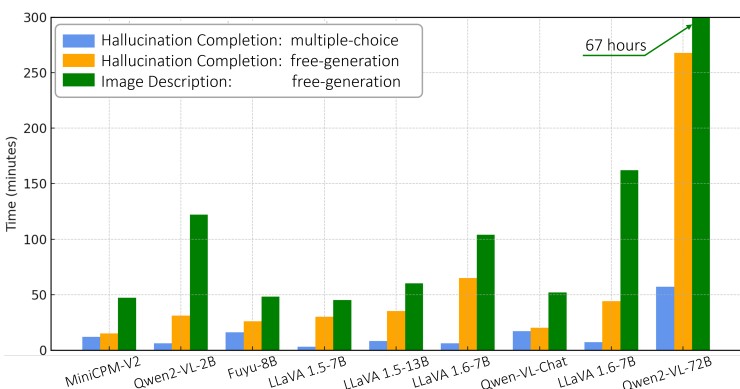

Figure 3: Comparison of evaluation times under different settings and MLLMs. Our proposed multiple-choice hallucination completion task is significantly faster than other(existing) setups, especially for large models. We measure the time taken to evaluate 1,000 image-text pairs under three different evaluation settings. Only the time that MLLMs take to generate text is measured without considering the time for evaluations by other LLM evaluators. All MLLMs are tested on one A100 except LLaVA 1.6-34B and Qwen2-VL-72B on an H100.

generating descriptions from scratch may be excessively long and impractical. The multiple-choice format of Longhalqa, while preserving evaluation effectiveness, significantly improves evaluation efficiency and facilitates future expansion of evaluation data in both scale and diversity.

## 7 CONCLUSION

This paper presents LongHalQA, a novel benchmark with long-context data for evaluating MLLMs' level of hallucinations in more practical scenarios. LongHalQA consists of 6.4k question-answer pairs with long-context data covering 12 types of hallucination. It features two multiple-choice tasks: hallucination recognition and hallucination completion, implementing both discriminative and generative hallucination evaluation in one unified format. It also offers additional assessments of the causes of hallucinations without involving LLM evaluators as in existing benchmarks. We also propose an automated pipeline for generating long-context hallucination data. Extensive tests reveal the constraints of existing MLLMs in handling long-context hallucinations, showing the necessity for more research on robust MLLMs with respect to long-context hallucinations.

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

## A  APPENDIX

## B  HALLUCINATION TYPES IN LONGHALQA

We present the definitions of different types of hallucinations as follows. Some examples are shown in Fig. 4 and Fig. 5.

H1 **(Non) Existent Objects.** The described objects do not exist in the given image.

H2 **Object Attributes.** The appearances (shape, pattern, etc.), types, or other attributes of objects are incorrectly described.

H3 **Object Color.** The colors of objects are incorrectly described. (Due to the high number of hallucinations of object color descriptions, we list "object color" separately from object attributes.)

H4 **Object States.** The states of objects, such as movement, orientation, the actions of the person, etc, are incorrectly described.

H5 **Number of Objects.** The number of objects is incorrectly stated.

H6 **Object Locations.** The locations of objects in the image are incorrectly described.

H7 **Object Relationships.** The relations or relative positions between two or multiple objects are incorrectly described.

H8 **Text/Sign Meaning.** The text in the image is wrongly discerned, or the meanings of signs, such as street signs, advertisements, price tags, etc, are incorrectly described.

H9 **Environment Description.** Wrong descriptions or adjectives of the environment or location, for example, indoors, outdoors, rural, urban, bookstore, food market, etc.

H10 **Background Description.** The descriptions of activities, scenes, objects, etc., in the background of the image are hallucinatory. Examples of descriptions include the presence of mountains, buildings, skyscrapers people in the background, or the description of no visible people in the background.

H11 **Time.** The incorrect description of the time of day, night, etc., in the image, or arbitrary judgment of the time without clear evidence.

H12 **Weather.** Incorrect descriptions of weather or sky conditions.

## C  DETAILS OF LONGHALLGEN

We present the prompts for different steps in LongHallGen in Fig. 6, Fig. 7, Fig. 8, and Fig. 9.

## D  CHAIN-OF-THOUGHT PROMPT FOR EVALUATION

We examine the impact of Chain-of-Thought on LongHalQA. We append additional prompts as shown in Fig. 10 before the questions for the hallucination discrimination task and the hallucination completion task.

H1

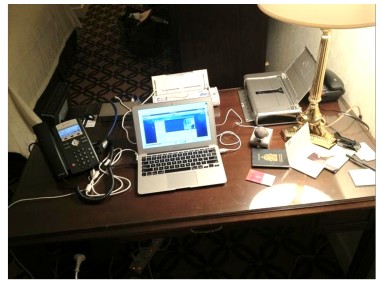

The image showcases a wooden desk in a room scattered with various electronic devices and objects. A laptop with an open page is centrally placed on the desk. To the right of the laptop, there is a second closed laptop and a printer. On the left, there is a desk phone connected to a cord. Multiple cards, notepads, and a pen are also visible on the desk. There is a lamp with a golden base and a cream-colored shade to the right of the desk. The floor has a geometric pattern, and a portion of a bed can be seen on the far left.

H2

The image shows four musicians performing onstage with a red curtain as the background. From left to right: a man seen from the back wearing a dark suit is playing a grand piano, a man with dreadlock hair is playing the double bass, another man in a dark jacket and brown pants is playing the saxophone, and a man in glasses wearing a dark suit is playing a drum set labeled 'YAMAHA'. The ambience suggests an intimate live jazz performance with dim, warm lighting.

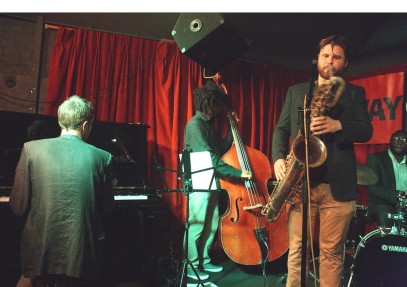

H3

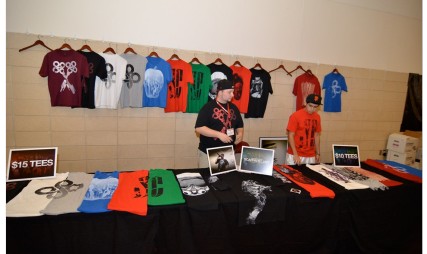

The image displays a merchandise booth with a variety of t-shirts for sale. Hanging on a makeshift clothesline above the booth are twelve t-shirts in different colors, including maroon, black, grey, blue, orange, red, and yellow, each with different graphic designs. Below, on the booth table, more t-shirts and folded clothing items are neatly arranged in rows, with price signs indicating "$15 TEES" and "$10 TEES." Two male individuals appear to be attending the booth, …….

H4

In the image, a man stands to the left side in what appears to be a kitchen or a kitchenette within an office environment. He is dressed in a blue buttoned-down shirt, grey trousers, and wears a belt. The man's expression is neutral, and his hands are in his pockets. In front of him is a table laden with various food items, likely set up for a meal or a gathering. The table presents a variety of dishes including what appears to be pies, bowls filled with what might be berries or salads, plates with pastries or rolls, and at least one large silver mixing bowl. The table also holds napkins ……

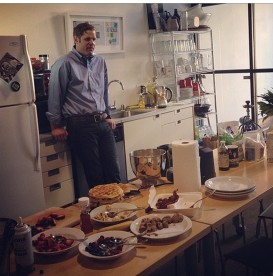

H5

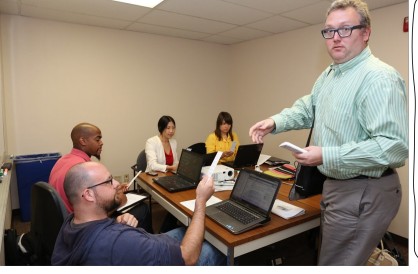

The image depicts an office environment. There are four people in the image, with one standing man and three seated individuals, two men, and a women. The standing man, wearing glasses and a green striped shirt, is handing a piece of paper to the seated man with a beard, who is wearing a dark shirt and is reaching out to take the paper. The second seated man, wearing a pink shirt, is looking at the paper, while a seated woman, in a white jacket and red top……

H6

The image showcases an urban city square with a mixture of classic European architecture. There are multi-story buildings with different facades, colors, and architectural details. To the left, there is a market area with various tents and a concentration of people, suggesting a bustling atmosphere. The foreground features construction with visible orange cones and barriers, indicating ongoing development or maintenance. There are several trees scattered ……

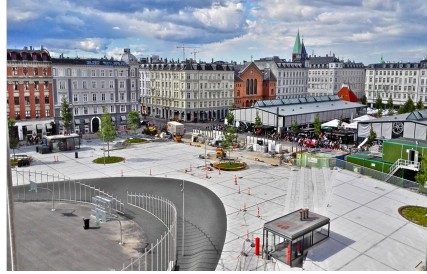

Figure 4: Visualizations of hallucination types from H1 to H6.

H7

The image shows a section of a highway during the daytime with clear skies. Road signage indicates an upcoming exit towards the right - Exit 27 for Route 2 East towards Boston. Above it is a directional sign for 91 North toward Brattleboro VT. There are four visible vehicles. The closer ones are a dark blue SUV and a dark-colored SUV, both heading towards Exit 27 on the rightmost lane. In the distance, a white commercial truck and a blue car are driving on the left lanes to Brattleboro VT. Trees with autumn foliage ......

The image showcases two red Los Angeles Fire Department SUVs parked on a dirt ground, with the left vehicle labeled "B 15" and the right labeled "EM 10". There are visible license plates on both vehicles. In the background, above the SUV, a helicopter is in flight, against a background of a clear sky and a field with green grass. There's a fire engine partially visible in the background to the left.

H8

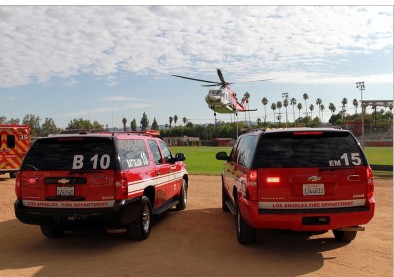

H9

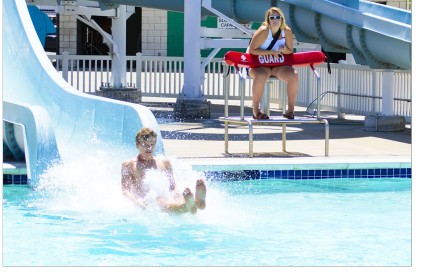

The image captures an indoor water park scene where a person is emerging from a blue water slide with water splashing around, presumably into a swimming pool. To the right, there is a lifeguard seated on an elevated chair. The lifeguard is wearing sunglasses and holding onto a red rescue tube marked with the word "GUARD.". The swimming pool is bordered by blue tiles. Behind the lifeguard, there is a circle of white railings enclosing the area.......

The image features a vibrant red offshore support vessel named "SKANDI CARLA" docked in calm blue waters, with the name and logo "DOF" prominently displayed on its hull. A smaller black and white tugboat is positioned parallel to the support vessel's starboard side, slightly towards its stern. Both vessels are..... The background features mountains covered in thick fog and buildings, including a multi-story brown building with glass windows. The sky is partly cloudy.

H10

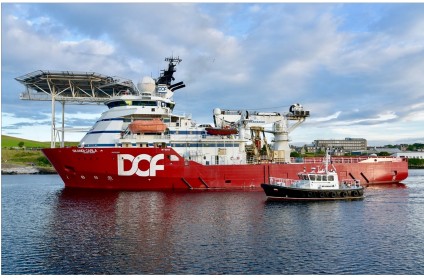

H11

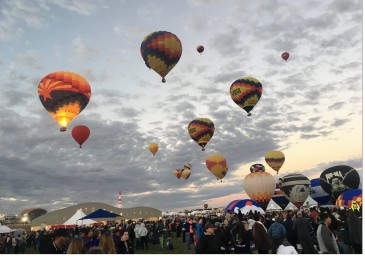

The image captures a lively hot air balloon festival at midday with numerous colorful balloons in the sky and a crowd of spectators on the ground. More than a dozen balloons are visible, adorned with various patterns and designs, floating against a backdrop of a cloudy sky. The scene is busy with activity on the ground, where people are gathered in groups, with some walking and others standing, enjoying the view. Tents and booths are set up, and there is a large white building in the background.

The image depicts a commercial airplane from the airline Southwest on a tarmac at an airport. This blue aircraft, marked by its distinct logo and the colors yellow, red, and orange on the tail, is possibly preparing for takeoff or taxiing on the runway...... marker indicating "F8" and an arrow suggesting a route. The environment suggests a typically busy airport scene with various facilities and structures in the background, likely an urban setting based on the buildings, and overcast weather conditions with visible clouds in the sky.

H12

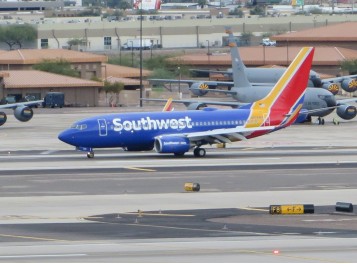

Figure 5: Visualizations of hallucination types from H7 to H12.

## 🔓 Prompt for Positive Data Generation

{ System Prompt: You are a powerful visual expert, a powerful image captioner, and an AI visual assistant who can understand and analyze image content and construct relevant data.}
{ Prompt for Specific Data Format }. Content that might be mentioned include { Possible Content }, etc. Only describing the content one can determine confidently from the image. Minimize aesthetic descriptions as much as possible.
The following is example template for your response:
{ Response Template }

### Prompt for Object-level Description

{ System Prompt } You are a powerful visual expert, a powerful image captioner, and an AI visual assistant who can understand and analyze image content and construct relevant data. Construct ten positive descriptions each for a specific object or specific objects in the image. Each description should be in one sentence. Content that might be mentioned for describing objects include object types, colors, states, actions, number of objects, precise object locations, texts or OCR results, relationships or relative positions between objects, etc. Only describing the content one can determine confidently from the image. Minimize aesthetic descriptions as much as possible. Do not describe what you can not discover from the image.
The following is an example template for your response:
Object Descriptions:\n1.\n2.\n3.\n4.\n5.\n6.\n7.\n8.\n9.\n10.

### Prompt for Image-level Description

{ System Prompt } Create a detailed description to describe the content of the given image. Compress the description in one paragraph. Content that might be mentioned include object types, colors, states, actions, number of objects, precise object locations, texts or OCR results, relationships and relative positions between objects, environment, background, time, weather, etc. Only describing the content one can determine confidently from the image. Minimize aesthetic descriptions as much as possible. Do not describe what you can not discover from the image.
The following is an example template for your response:
Image Description:

### Prompt for Multi-round Conversation

{ System Prompt } Generate conversation data for a situation where a user and an assistant discuss the content and details of the image. The user's role is to ask various questions about the image. The role of the assistant is to respond to the user's questions. During the conversation, the user can ask multiple progressive questions about certain content or objects in the assistant's previous answers to inquire about their details from general to specific. Except when starting a new topic, users should try to ask specific questions and avoid asking overly broad questions. Contents that can be discussed in the conversations include types, colors, states, and actions of objects, the number of objects, object locations, text or OCR results, doublechecked relative positions or relationships between objects, environment, background, time, weather, etc. The conversation should be natural, continuous, and logical. Ensure that the conversation remains focused on the content one can determine confidently from the image. Minimize aesthetic descriptions and keep the interaction informative and engaging.
Then, edit the assistant's answer in the generated conversation data and deliberately introduce one misleading error that doesn't match the image's content, leaving most of the content unchanged. You can modify object states, actions, number of objects, relationships or relative positions between objects, environment, background, time, etc., or introduce plausible objects that may exist according to the scene of the image. Explain the error in the modified conversation.
The following is an example template for generated data.
Original Conversation:
User:[Start the conversation by asking a question about the image.]
Assistant:[Answer the user's question based on the image content.]
User:[Continue the conversation by 1)seeking clarification on information in the previous answer; 2)asking additional questions based on previous questions; 3) asking new questions.]
Assistant:[Answer the user's question. Maintain a fluid and coherent dialogue flow.]....
[continue the conversation until the image is fully discussed.]
Modified Conversation:[Edit the original conversation and deliberately introduce one misleading error that doesn't match the image's content, leaving most of the content unchanged.]
Explanation:[Explain the error in the modified conversation.]

Figure 6: Prompt for generating positive data. We prompt GPT-4V to generate multiple conversation data to make it respond stably.

**👤 Prompt for Hallucination Check**

{ System Prompt: You are a powerful visual expert, a powerful image captioner, and an AI visual assistant who can understand and analyze image content, construct and analyze relevant data.}
{ Prompt for Specific Data Format } Pay special attention to check { Possible Content }.
Following is the { Data Format } :
{ Generated Data }
Following is an example template for your response:
{ Response Template }

**Prompt for Object-level Description**

{ System Prompt } You will be given 10 sentences describing the objects in the image. Check each sentence of description carefully to see if any content does not match the image content. If the description exactly matches the contents in the image, output [Match]. If anything in the description does not fit the image, output [Do not Match] and explain. Pay attention to the content of the description including objects' types, colors, states, actions, number of objects, precise object locations, texts or OCR results, relationships and relative positions between objects, environment, background, time, weather, etc.
The following are the descriptions of objects:\n{}
Following is an example answer template:
1.Origin Sentence:[The 1st description]. Result:[Match] / [Do not match][Explanations if it does not match.]
2.Origin Sentence:[The 2st description]. Result:[Match] / [Do not match][Explanations if it does not match.].......

**Prompt for Image-level Description**

{ System Prompt } Check each sentence in the following description carefully to see if any content does not match the image content. If the sentence exactly matches the image content, output [Match]. If there is anything in the sentence that does not fit the image, output [Do not Match] and give an explanation. Pay attention to the content of the description including objects' types, colors, states, actions, number of objects, precise object locations, texts or OCR results, relationships and relative positions between objects, environment, background, time, weather, etc.
The following is the description:\n{}
Following is an example template for your response:
1.Origin Sentence:[The 1st sentence]. Result:[Match] / [Do not match][Explanations if it does not match.]
2.Origin Sentence:[The 2st sentence]. Result:[Match] / [Do not match][Explanations if it does not match.].......

**Prompt for Multi-round Conversation**

{ System Prompt } You will be given a conversation between the user and the assistant. Carefully check whether there is any content in each sentence of each assistant's answer that does not match the image content. If the sentence exactly matches the image content, output [Match]. If there is anything in the sentence that does not fit the image, output [Do not Match] and give an explanation. First, analyze each sentence and find its mentioned objects and corresponding content, such as objects' types, colors, states, actions, number of objects, precise object locations, texts or OCR results, relationships and relative positions between objects, environment, background, time, weather, etc. Then, check whether each mentioned object exists in the image. If it exists, then check the correctness of each piece of content corresponding to that object.
The following is the conversation:\n{}
Following is an example answer template:
1.1.Origin Sentence:[1st sentence in 1st answer]. Result:[Match] / [Do not match][Explanations if it does not match.]
1.2.Origin Sentence:[2nd sentence in 1st answer]. Result:[Match]/[Do not match][Explanations if it does not match.]
.......
3.1.Origin Sentence:[The 1nd sentence in the 3st answer] .......

Figure 7: Prompt for *Hallucination Check* in LongHallGen.

### 👤 Prompt for Hallucination-Explanation Pair Generation

{ System Prompt: You are a powerful visual expert who can understand and analyze the content of images and construct relevant data.}

Prompt for Hallucinatory Data:

You will be given { Data Format } of the image, and an analysis about incorrect sentences in the { Data Format } that do not match the image content and corresponding reasons. Your task is to 1) generate the correct { Data Format } based on the above content. 2) Select at most three errors and then modify the { Data Format } for each selected error (correct other errors in the { Data Format } so that it only contains this one error). Output the correct { Data Format }, the { Data Format } corresponding to each error, and the explanation of the error. Pay attention to maintaining the logic and coherence of { Data Format } when making modifications.

Prompt for Non-Hallucinatory Data:

You will be given { Data Format } of the image. Your task is to edit the { Data Format } and deliberately introduce one misleading error that doesn't match the image's content, leaving most of the content unchanged. You can modify { Types of Hallucinations }. Pay attention to maintaining the logic and coherence of description when making modifications. Output the original description, the modified description and the explanation of the introduced error. Do not illustrate the modification, only explain the error in the description

Following is the description: { Generated Data }
Following is the analysis: { Hallucination Analysis }
Following is an example template for the response: { Response Template }

### Prompt for Image-level Description

{ System Prompt }

Prompt for Hallucinatory Data:

......Following is an example template for your response:

Correct Description:[Description without errors.]

Explanation:[Briefly summarize the content of the correct description in one sentence to suggest the description matches the image content.]

Incorrect Description:[Description with one error preserved.]

Wrong Sentence in Description :[the incorrect sentence in the above modified description.]

Explanation:[Explain the error in the description.]

......[Repeat the process to generate incorrect descriptions for each error in the description.]

Prompt for Non-Hallucinatory Data:

Original Description:[Description without errors.]

Explanation:[Briefly summarize the content of the correct description in one sentence to suggest the description matches the image content.]

Incorrect Description:[Description with one error introduced.]

Wrong Sentence in Description :[the incorrect sentence in the above modified description]

Explanation:[Explain the error in the description.]

### Prompt for Multi-round Conversation

{ System Prompt }

Prompt for Hallucinatory Data:

......Following is an example template for your response:

Correct conversation:[Complete conversation without errors.]

Explanation:[Briefly summarize the content of the correct conversation in one sentence to suggest the conversation matches the image content.]

Incorrect Conversation 1:[The modified complete conversation with one error preserved.]

Wrong Sentence in Conversation:[the incorrect sentence in the above modified conversation]

Explanation:[Explain the error in this incorrect conversation.]

Incorrect Conversation 2: ...

Prompt for Non-Hallucinatory Data:

Original Conversation:[Conversation without errors.]

Explanation:[Briefly summarize the content of the conversation in one sentence to suggest the conversation matches the image content.]

Incorrect Conversation:[Conversation with one error introduced.]

Wrong Sentence in Conversation:[the incorrect sentence in the above modified conversation]

Explanation:[explanation of the error in the conversation.]

Figure 8: Prompt for *Hallucination-Explanation Pair Generation* in LongHallGen.

### 👤 Prompt for Constructing the Hallucination Discrimination task

{ System Prompt: You are a powerful visual expert who can understand and analyze the content of images and construct relevant data.}
You will be given { Data Format }, and an answer of whether the { Data Format } match the image content and corresponding reasons if incorrect. Your task is to construct multiple answer options for the question { Question }. The given answer should be paraphrased as one of the options and the correct option, then add \"(correct)\" as a tag after this correct option. Options are ordered by \"A.\", \"B.\", \"C.\", and \"D.\". Each answer option comprises of a \"Yes\" or \"No\" and an explanation. One of the options should begin with yes, and give an statement indicating that { Data Format } match the image content. The other three options should start with \"No\" and point out a possible error in { Data Format }. Options other than the correct one should contain errors. Errors can be of two types: 1). the incorrect \"yes\" or \"no\" compared to the correct option; 2) the given explanation is incorrect. Do not construct explanations by indicating that the { Data Format } does not mention something in the image. For all answers, the explanations given must be logical, appropriate to the question, relevant to the image content, and be compressed in one sentence.
Following is the { Data Format }:\n{}
Following is the answer:\n{}
Following is an example template for your output.
Question: { Question }
Answer Options:\nA.\nB.\nC.\nD.\n

### 👤 Prompt for Constructing the Hallucination Completion task

{ System Prompt: You are a powerful visual expert who can understand and analyze the content of images and construct relevant data.}
You will receive { Data Format }, and an analysis of why and which sentence of { Data Format } do not match the content of the image. Your task is to create a continuation task. First, output the text of the { Data Format } before the incorrect sentence. Then, provide four continuation options: one is the original incorrect sentence, another is the corrected sentence based on the analysis, the third option introduces a potential error by modifying the correct sentence, and the last option is a sentence containing an error that you write based on the preceding text and the image content. Add \"(correct)\" as a tag after the option of the correct sentence. Options are ordered by \"A.\", \"B.\", \"C.\", and \"D.\".
Following is the { Data Format }:\n{}
Following is the wrong sentence in the { Data Format }: \n{}
Following is the analysis: \n{}
Following is an example template for your output.
Question: Continue the following { Data Format } about the image.
Answer Options:\nA.\nB.\nC.\nD.\n

Figure 9: Prompt for *Question and Answer Generation* for the hallucination discrimination task and the hallucination completion task, respectively.

### 👤 Modified Prompt for Hallucination Discrimination

You are a powerful AI assistant who can understand the image content and answer relative questions. You will be given a text about the image content. Your task is to check whether the text data match the image content. Please think step by step. First, analyze the image content including objects' types, colors, states, actions, number of objects, precise object locations, texts or OCR results, relationships and relative positions between objects, environment, background, time, weather, etc. Then, carefully check each sentence in the text data based on the above analysis of image content.

### 👤 Modified Prompt for Hallucination Completion

You are a powerful AI assistant who can understand the image content. You will be given a text about the image content. Your task is to continue the text. When continue writing, Pay attention to the image content including objects' types, colors, states, actions, number of objects, precise object locations, texts or OCR results, relationships and relative positions between objects, environment, background, time, weather, etc. Be careful to avoid from responding text that contains hallucinations or error.

Figure 10: Modified prompts that involve Chain-of-Thought for the hallucination discrimination task and the hallucination completion task.

