# OpenReview forum: "LongHalQA: Long-Context Hallucination Evaluation for MultiModal Large Language Models"
_ICLR.cc/2025/Conference — Submitted to ICLR 2025_

### Official Review · Reviewer_Xq4o · 2024-10-18

**Soundness:** 2
**Presentation:** 2
**Contribution:** 2
**Rating:** 5
**Confidence:** 5

**Summary:**

The paper addresses the issue of hallucination in multimodal large language models (MLLMs), where generated text doesn't match the input image. To solve problems with existing benchmarks, the authors propose LongHalQA, a new benchmark with 6,000 complex hallucinatory texts that mimic real-world scenarios. It introduces two tasks: hallucination discrimination and hallucination completion, which combine both discriminative and generative evaluations into a single multiple-choice format. This approach avoids the need for LLM evaluators, making the evaluation process more reliable and efficient. The paper also presents a new pipeline for creating complex hallucination benchmarks and provides experiments showing that recent MLLMs struggle with long, complex text.

**Strengths:**

1. LongHalQA addresses the limitations of previous benchmarks by creating a comprehensive dataset of hallucination text that mirrors real-world scenarios, providing a more accurate and complex testing environment for MLLMs.

2. By eliminating the need for LLM evaluators, the benchmark ensures more stable and reliable results, avoiding the randomness and computational intensity associated with LLM-based evaluations.

3. The combination of both discriminative and generative evaluation tasks in a multiple-choice format allows for a holistic assessment of MLLM performance in handling hallucinations, making the evaluation process more efficient.

**Weaknesses:**

1. How to evaluate model with the Hallucination Completion task?  What is the prefix text for evaluation? Is it the first word?
2. Why the Hallucination Completion can be seen as generative evaluation? The multi-choice question still is discriminative question.
3.  “then analyze and filter them based on dataset annotations and GroundingDINO”: how did authors analyze and filter?
4. Lack of comprehensive survey of hallucination on Large Vision-Language Models.
[1] Object hallucination in image captioning
[2] Halle-switch: Rethinking and controlling object existence hallucinations in large vision language models for detailed caption
[3] FaithScore: Fine-grained Evaluations of Hallucinations in Large Vision-Language Models
[4] Analyzing and mitigating object hallucination in large vision-language models
[5] FGAIF: Aligning Large Vision-Language Models with Fine-grained AI Feedback
5. The proposed LLM-free hallucination benchmark does not offer significant advantages, as the approach still requires various tools, LVLMs, and manual verification, leading to low efficiency.
6. The benchmark has not demonstrated greater reliability compared to existing ones, such as through experimental validation.

**Questions:**

See weakness

---

> ### Author Response · Authors · 2024-11-20
>
> Thanks very much for your insightful comments. They are very helpful in improving our paper. In the following, we first state your comments and follow with our point-to-point response.
>
> > 1. How to evaluate model with the Hallucination Completion task? What is the prefix text for evaluation? Is it the first word?
>
> Here is an intact example of our query for the hallucination completion task:
> ```
> Continue the following description of the image.
> The image showcases a wooden desk in a room scattered with various electronic devices and objects. A laptop with an open page is centrally placed on the desk. On the left, there is a desk phone connected to a cord. To the right of the laptop, there is a second desk phone and a printer. Multiple cards, notepads,
> A. a mobile phone charger, and a pen are also visible on the desk.
> B. and a pen are also visible on the desk.
> C. and markers are also visible on the desk.
> D. a pair of earbud headphones, and a pen are also visible on the desk.
> Answer with the option's letter from the given choices directly.
> ```
> We will include more question examples from LongHalQA in the appendix for clarity.
>
> > 2. Why the Hallucination Completion can be seen as generative evaluation? The multi-choice question still is discriminative question.
>
> Thank you for this insightful question. We believe that the Hallucination Completion task in LongHalQA provides a structured way to quantify MLLMs' generative quality, which complements existing generative benchmarks by addressing their limitations in terms of data and hallucination diversity, efficiency, and evaluation randomness. By providing multiple plausible options, we simulate the generative scenario of MLLMs within a framework that is easier to compare and score. As demonstrated in Table 8, the rankings of hallucination levels under the MCQ completion task are largely consistent with those from open-ended generative scenarios, demonstrating that the proposed MCQ task is able to capture the generative capabilities of MLLMs.
>
> In addition, the MCQ hallucination completion task effectively simulates the sentence-level beam search generation process of MLLMs. Specifically, given the pre-generated text, MLLMs are required to generate multiple candidate options and then select the most reasonable, hallucination-free option as the completion result. Under such a scenario, if a model consistently selects non-hallucinatory candidates for each generated sentence, it can be considered to have a relatively low level of generative hallucinations.
>
> > 3. “then analyze and filter them based on dataset annotations and GroundingDINO”: how did authors analyze and filter?
>
> We leverage GroundingDINO to remove those inaccurate annotated boxes labeled as "crowd" and remove images that lack sufficient complexity or richness of content~(such as containing less than five objects). We will update the paper to make it clearer.
>
> > 4. Lack of comprehensive survey of hallucination on Large Vision-Language Models. [1] Object hallucination in image captioning [2] Halle-switch: Rethinking and controlling object existence hallucinations in large vision language models for detailed caption [3] FaithScore: Fine-grained Evaluations of Hallucinations in Large Vision-Language Models [4] Analyzing and mitigating object hallucination in large vision-language models [5] FGAIF: Aligning Large Vision-Language Models with Fine-grained AI Feedback
>
> Thank you for your suggestion. We will review the suggested work carefully in the Related Work section in the updated paper.
>
> > 5. The proposed LLM-free hallucination benchmark does not offer significant advantages, as the approach still requires various tools, LVLMs, and manual verification, leading to low efficiency.
>
> We would like to clarify that no additional tools, LVLMs, or human verification are required during the evaluation process. These costs are only involved while constructing the LongHalQA benchmark itself. For evaluation, we directly test models using binary (yes/no) or multiple-choice questions. As illustrated in Figure 3, our LongHalQA provides far more efficient evaluation than generative benchmarks that require waiting for MLLMs to generate content.

---

> ### Author Response · Authors · 2024-11-20
>
> > 6. The benchmark has not demonstrated greater reliability compared to existing ones, such as through experimental validation.
>
> Thank you for raising this point. To demonstrate LongHalQA's reliability, we evaluate several hallucination mitigation methods on LongHalQA and compare the evaluation results with those over existing benchmarks, including POPE (for object hallucinations), MMHal (comprehensive questions about image content with answers scored by GPT), and MRHal (hallucinations for multi-round dialogue evaluated by GPT). The improvements of evaluated methods over LongHalQA (for object descriptions under the hallucination discrimination task, image descriptions, and multi-round conversations under the hallucination completion task) are generally consistent with those achieved over POPE, MMHal, and MRHal, demonstrating the reliability of LongHalQA in MLLM evaluations.
>
> Furthermore, several points can be observed through a comprehensive analysis of the evaluation results with the LongHalQA:
>
> (1) Most methods use image description tasks to construct preference data, and thus we observe that most methods significantly improve over the baseline for image description data in the completion task.
>
> (2) This preference optimization based on image description also effectively improves performances in discriminating hallucinations of object description, consistent with the gains observed on prior benchmarks such as POPE. However, these methods showed limited improvements for discrimination tasks in long-context settings, which involve detailed image descriptions and multi-round conversation. When models are required to identify the reasons for hallucinations correctly, most of their performances for hallucination discrimination tasks under multiple-choice question (MCQ) settings decrease.
>
> (3) The preference optimization based on the image description task yields limited benefits for multimodal conversational capabilities. For example, CSR and POVID result in performance drops to around 20\% on discrimination tasks for the conversation. We found this is due to the impaired instruction-following capabilities, where they fail to process long-context queries and correctly output the option letter. Furthermore, the improvements in hallucination completion tasks for conversation data are also limited compared to their baseline models.
>
> These findings highlight the current limitations of preference-based hallucination mitigation methods in handling long-context scenarios, as well as the lack of diversity in tasks and data formats. Increasing the diversity and length of constructed preference data might help address these issues. Overall, these findings underscore the distinctive value of LongHalQA in evaluating complex hallucinations in MLLMs within long-context scenarios.
>
> Table 1: Experiments of methods for mitigating hallucinations of MLLMs on existing benchmarks.
> | Method            | POPE  | MMHal | MRHal |
> |:------------|:------|:------|:------|
> | LLaVA-1.5-7B   | 85.9  | 2.36  | 3.38  |
> | + CSR            | 87.1  |   -   |   -   |
> | + POVID          | 86.9  | 2.69  | 3.46  |
> | + RLAIF          |   -   | 3.06  |   -   |
> | LLaVA-1.5-13B | 85.9  |   -   | 3.58  |
> | + LLaVA-RLHF     |   -   |   -   |   -   |
> | Qwen-VL-Chat  | 87.1 | 2.89  | 3.71  |
> | + Silkie         |   -   | 3.02  | 3.71  |
> | Muffin-13B    |   -   |   -   |   -   |
> | + RLHF-V         |   -   | 2.45  | 2.54  |
>
> Table 2: Experiments on methods for mitigating hallucinations of MLLMs on LongHalQA. ''Obj.'', ''Des.'' and ''Con.'' are the data formats of object description, detailed image description, and multi-round conversations in LongHalQA. Columns 2-6 are the results of the hallucination discrimination task, and columns 7-8 are the results of the hallucination completion task.
>
> |                       |        |    Hallucination       |  Discrimination  | | | Hallucination | Completion |
> |:------------|:------:|:------:|:------:|:------:|:------:|:------:|:------:|
> | **Methods**           |  **Obj.(Binary)**  |  **Des.(Binary)**  |  **Con.(Binary)**  |  **Des.(MCQ)**  |  **Con.(MCQ)**  | **Des.** | **Con.** |
> | LLaVA-1.5-7B   | 45.18  | 36.59  | 33.79  | 37.17  | 32.92  | 32.80  | 39.37  |
> | + CSR            | 44.52  | 36.59  | 31.30  | 36.66  | 20.16  | 32.86  | 36.54  |
> | + POVID          | 45.69  | 36.66  | 32.54  | 37.03  | 21.20  | 36.25  | 40.86  |
> | + RLAIF          | 60.29  | 36.88  | 35.79  | 35.28  | 30.17  | 36.48  | 44.02  |
> | LLaVA-1.5-13B | 52.70  | 36.95  | 35.85  | 45.99  | 41.21  | 31.53  | 43.62  |
> | + LLaVA-RLHF     | 48.83  | 36.66  | 36.91  | 41.11  | 37.78  | 37.17  | 44.02  |
> | Qwen-VL-Chat  | 58.69  | 36.66  | 34.29  | 37.97  | 36.10  | 33.14  | 40.00  |
> | + Silkie         | 62.19  | 36.95  | 35.22  | 37.54  | 35.29  | 37.17  | 40.63  |
> | Muffin-13B    | 46.50  | 52.99  | 67.46  | 38.85  | 30.92  | 17.15  | 26.46  |
> | + RLHF-V         | 60.29  | 38.19  | 40.90  | 38.12  | 25.62  | 31.99  | 31.02  |

---

> > ### Comment · Reviewer_Xq4o · 2024-11-22
> >
> > Thanks for your response. The response still doesn't address my concerns, such as considering the multi-choice question as a generative evaluation but not a discriminative evaluation, the advantage of efficiency, and the lack of verification of the benchmark (such as human correlation). Therefore, I will keep my score.

---

> > > ### Author Response · Authors · 2024-11-25
> > >
> > > Thank you for your response. Regarding your points about generative and discriminative evaluations, efficiency issues, and benchmark validation, we reply to each of these aspects below and hope to address your concerns.
> > >
> > > >1. The consistency between MCQ evaluation and free-form generative evaluation and the advantage efficiency.
> > >
> > > LongHalQA includes two tasks: hallucination discrimination and hallucination completion. Our hallucination completion task adopts a discriminative multiple-choice format to simulate generative evaluation, addressing randomness and reliance on external MLLMs and reducing evaluation costs. The task prompt we designed, "Continue the following description of the image." is also closely aligned with the generative evaluation prompt, "Describe the given image," rather than the traditional hallucination discrimination benchmarks, which typically focus on binary yes-or-no questions about the object existence or whether statements are true.
> > >
> > > We believe that the discriminative MCQ format and free-form generative evaluation each have their strengths and can complement one another. The advantage of generative evaluation lies in allowing direct access to the model's outputs. However, it has high evaluation costs, depends on external models, and struggles to reliably assess complex hallucinations. In contrast, the MCQ format could directly leverage those challenging visual content prone to hallucinations for MLLM evaluation. MCQ also reduces evaluation costs, eliminates the need for external models, and provides more definitive assessments.
> > >
> > > Table 8 in our paper demonstrates the consistency between the MCQ and generative evaluation in ranking MLLMs. We also conduct the following experiment using the image description task to rank MLLMs by their hallucination levels and detailedness, with more hallucination mitigation methods evaluated in the table. Our MCQ evaluation aligns closely with this generative evaluation based on the typical image description task.
> > >
> > > In terms of efficiency advantages, as shown in Figure 3, the MCQ format is significantly more efficient than the original generative evaluation by free-form completion in Table 8 or the image description task in the table below. Moreover, Figure 3 only reflects generation time and does not account for the additional evaluation time for the generative evaluations. The MCQ format could directly output accuracy scores without extra evaluation steps, resulting in even greater efficiency.
> > >
> > > Table 1. Comparison of MCQ evaluation and generative evaluation with the image description task. The last two columns are the performance and ranking of the MCQ hallucination completion task with image description data.
> > > |               |                 |    Free     |    Generated     |  Description  |                |                |     Hall     |   Completion   |
> > > |:--------------|:---------------:|:-----------:|:----------------:|:-------------:|:--------------:|:--------------:|:------------:|:--------------:|
> > > | **Methods**     | **Number of Words** | **Detail Rank** | **Detail Score** | **Hall Rank** |**Hall Score**| **Avg. Rank**  | **MCQ Rank** | **Desc.(MCQ)** |
> > > | MiniCPM-V2    |      135.3      |      2      |       6.45       |       3       |      6.66      |       2        |      2       |     44.07      |
> > > | Qwen2-VL-2B   |      107.6      |      3      |       6.34       |       2       |      6.78      |       1        |      1       |     47.18      |
> > > | LLaVA 1.6-7B  |      177.4      |      1      |       6.52       |       6       |      5.83      |       4        |      3       |     39.47      |
> > > | LLaVA 1.5-7B  |      104.8      |      8      |       5.46       |      10       |      5.38      |       9        |      9       |     32.80      |
> > > | + POVID       |      95.5       |     10      |       5.34       |       5       |      5.98      |       6        |      7       |     36.25      |
> > > | + RLAIL       |      104.1      |      7      |       5.52       |       1       |      6.80      |       4        |      6       |     36.48      |
> > > | LLaVA 1.5-13B |      102.9      |     11      |       5.33       |      11       |      5.25      |       11       |      11      |     31.53      |
> > > | + LLaVA-RLHF  |      116.5      |      6      |       5.54       |       7       |      5.78      |       6        |      4       |     37.17      |
> > > | Qwen-VL-Chat  |      97.3       |      5      |       5.70       |       9       |      5.55      |       8        |      8       |     33.14      |
> > > | + Silkie      |      99.2       |      4      |       6.23       |       4       |      6.34      |       3        |      4       |     37.17      |
> > > | Muffin        |      100.8      |      9      |       5.45       |      11       |      5.25      |       10       |      12      |     17.15      |
> > > | +RLHF-V       |      51.3       |     12      |       4.33       |       8       |      5.60      |       12       |     10       |    31.99     |

---

> ### Author Response · Authors · 2024-11-25
>
> We adopt the following process for generative evaluation of MLLM hallucination with the help of GPT in the above comment.
>
> ```
> For each MLLM, we input the prompt: "Describe the given image in detail." We then feed the image, the revised description from LongHalQA, and object annotations from Objects365 to GPT，and ask GPT to score the MLLMs generated descriptions in terms of detail level and hallucination degree. Both scores range from 0 to 10, with higher scores indicating better performance(more detailed and less hallucination content).
> ```
>
> > 2. Verification of the benchmark
>
> To further validate the benchmark, we randomly selected 100 questions each from the discrimination and completion tasks across three data types in LongHalQA. We then had human evaluators answer these questions, and the results are shown in the table below. Human evaluators achieve an accuracy of around 90% on most tasks, highlighting that there is still significant room for addressing MLLM hallucinations.
>
> In addition, Table 8 in the paper, along with the comparisons in the above comment for the comparison with the generative evaluation using GPT,  could further demonstrate the effectiveness of our benchmark.
>
> Table 1. Human verification of the LongHalQA benchmark. We randomly sample 100 questions from each task and data format and ask human evaluators to answer the questions. ''Object'', ''Desc.'' and ''Conv.'' denote object description, detailed image description, and multi-round conversation, respectively.
> | | Hall. | Discrimination | | |   Hall.   | Completion  |
> |:------------------:|:-----------------:|:-----------------:|:--------------:|:--------------:|:---------:|:-----------:|
> | **Object(Binary)** | **Desc.(Binary)** | **Conv.(Binary)** | **Desc.(MCQ)** | **Conv.(MCQ)** | **Desc.** |  **Conv.**  |
> |        93%         |        89%        |        82%        |      92%       |      89%       |   84%    |    89%     |

---

> ### Author Response · Authors · 2024-12-01
> **Looking forward to further discussion**
>
> We sincerely appreciate your insightful and valuable comments. We have further supplemented comparisons between free-form generation and our MCQ approach on the image description task to demonstrate the consistency between the MCQ and generative evaluation, as well as human validation results to prove the reliability of the LongHalQA benchmark. As the discussion phase is about to close, we kindly ask you to take a few minutes to review our responses. If our responses have clarified your concerns, we hope you might consider raising the score. We look forward to hearing from you about any further feedback or suggestions for improving our work.

---

> > ### Comment · Reviewer_Xq4o · 2024-12-03
> >
> > Thanks for your response. I will keep my score based on our discussion.

---

### Official Review · Reviewer_Nmef · 2024-10-30

**Soundness:** 2
**Presentation:** 3
**Contribution:** 2
**Rating:** 5
**Confidence:** 4

**Summary:**

This paper proposes a new MLLM hallucination benchmark consisting of both hallucination discrimination and hallucination completion questions. The author unifies both discriminative and generative hallucination evaluation into the form of multiple-choice question where models only have to decode one token as response. The results show the proposed benchmark is challenging for both open-source MLLMs in varying sizes and strong GPT-4o.

**Strengths:**

1. The proposed benchmark can contribute the further development of this field tackling and analyzing the hallucination of MLLMs.
2. The proposed unification of discriminative question and generative question largely saves the evaluation cost via reducing the decoding sequence length.

**Weaknesses:**

1. Experimental results in Table 8 do not suggest a strong consistency between generation accuracy and mcq accuracy. For example, Fuyu-8b and LLaVA 1.5-7b exhibits score difference -12.41 in mcq while -41.0 in generation. It is necessary to include more methods into consideration, especially thous proposed to tackling hallucination of MLLMs such as LLaVA-RLHF, RLHF-V, Silkie and POVID.
2. Hallucination pairs are generated by GPT-4V, which are prone to generate hallucinated visual description. The author have to explain how #317 controls the generation quality.

**Questions:**

1. It is known that [LLMs are non-robust multiple-choice selectors](https://arxiv.org/abs/2309.03882). How do you tackle this problem during constructing this benchmark?
2. #419 mentions the 'ranking-based accuracy` of Fuyu-8B, while I could not find the corresponding results in Table 4. It is a writing issue?

---

> ### Author Response · Authors · 2024-11-21
>
> Thank you for your constructive comments and suggestions, which are exceedingly helpful in improving our paper. Our point-to-point responses to your comments are listed below.
>
> > 1. Experimental results in Table 8 do not suggest a strong consistency between generation accuracy and MCQ accuracy. For example, Fuyu-8b and LLaVA 1.5-7b exhibits a score difference -12.41 in MCQ while -41.0 in generation. It is necessary to include more methods into consideration, especially those proposed for tackle the hallucination of MLLMs such as LLaVA-RLHF, RLHF-V, Silkie, and POVID.
>
> We would like to clarify two points regarding the performance gaps shown in Table 8. (1) First, the low performance of the Fuyu model is primarily due to it sometimes fails to follow instructions to continue the descriptions/conversation. Instead, it often generates content unrelated to the given image, leading to unsatisfied accuracy. (2) Second, in the LongHalQA setting, we treat hallucinations generated by GPT as potentially misleading visual content, and evaluate MLLMs to directly describe such challenging content. However, in free-generation scenarios, even with preceding context as guidance, MLLMs sometimes still avoid addressing these challenging content and instead describe only the simplest visual content. This tendency explains why the accuracy in free-generation settings is generally higher than the completion task accuracy in LongHalQA, as shown in Table 8.
>
> Additionally, based on your suggestion, we also evaluate the performance of the hallucination-mitigated methods in both the MCQ and free completion scenarios. However, we observe significant differences in GPT responded results for the same prompts compared to the results we obtained a few months ago, perhaps due to their recent updates. So we re-evaluate some models in Table 8 and summarized the results alongside those from the hallucination-mitigation methods in the table below.
>
> Most hallucination-mitigation methods performed reasonably well in both the free-generation and MCQ settings. POVID and Silkie consistently achieve improvements over their baseline models, LLaVA-1.5-7B and Qwen-VL-Chat. However, while these methods reduce hallucination, they also affect the models' output behaviors and instruction-following abilities. For instance, POVID and RLHF-V tend to produce outputs significantly shorter than those of other models. LLaVA-RLHF sometimes fail to continue the descriptions / conversation based on prompts, instead generating an entire new image descriptions, which led to lower Free Generation Accuracy. In contrast, the original models in Table 8 produced outputs with more consistent lengths. As a result, directly comparing these models in the free-generation setting might be less fair. Our MCQ-based hallucination completion task, however, avoids the issue of output length differences by testing the models’ tendency to generate hallucinations under the same challenging scenarios, enabling fairer evaluations.
>
> Table1: Comparison of Multi-Choice and Free-Generation settings on Hallucination Completion for hallucination-mitigation methods.
> | Accuracy       | Number of Generated Words | Free Generation Accuracy | MCQ Accuracy |
> |:---------------|:--------------------------:|:--------------------------:|:-------------:|
> | LLaVA 1.5-7B   | 25.68                   | 73.5                     | 36.08        |
> | LLaVA 1.6-7B   | 38.74                   | 84                       | 43.40        |
> | LLaVA 1.5-13B  | 32.80                   | 74                       | 37.58        |
> | Qwen-VL-Chat   | 32.00                   | 70                       | 36.57        |
> |----------------|--------------------------|--------------------------|-------------|
> | POVID          | 14.38                   | 78.75                    | 38.55        |
> | RLHF-V         | 8.73                    | 74.5                     | 31.50        |
> | LLaVA-RLHF     | 147.79                  | 56.75                    | 40.60        |
> | Silkie         | 37.27                   | 74.25                    | 38.90        |

---

> ### Author Response · Authors · 2024-11-21
>
> > 2. Hallucination pairs are generated by GPT-4V, which are prone to generate hallucinated visual description. The author have to explain how #317 controls the generation quality.
>
> Thank you for raising this question. We would like to clarify that this step does not require GPT to describe the image from scratch. Instead, GPT only needs to revise its previously generated data based on the sentence-level analysis results obtained in the previous hallucination-checking step to construct the hallucination pairs. Although GPT-4V does face significant challenges with multimodal hallucinations, it still demonstrates strong text-processing capability, which is what we primarily rely on for constructing hallucination pairs. We also conducted random checks on the hallucination pairs generated by GPT and found that they generally met our requirements in revising the hallucinatory texts.
>
> > 3. It is known that LLMs are non-robust multiple-choice selectors. How do you tackle this problem during constructing this benchmark?
>
> We have considered this fact in constructing our MCQ benchmark. As mentioned in L261, We randomly shuffle the orders of the four options for each MCQ to reduce the impact of option order.
>
> > 4. #419 mentions the 'ranking-based accuracy` of Fuyu-8B, while I could not find the corresponding results in Table 4. It is a writing issue?
>
> Thank you for pointing this out. This is indeed a writing error on our part. We will revise the text in the paper.

---

> > ### Comment · Reviewer_Nmef · 2024-11-23
> >
> > Thanks for the response and effort of conducting more evaluations. The results however do not address my concern about the consistency of MCQ evaluation and generative evaluation. I would change my score to 5 as the results seem to be non-robust on methods tackling multimodal hallucination.

---

> ### Author Response · Authors · 2024-11-23
>
> Thank you very much for your feedback.
>
> However, we want to argue that LongHalQA achieves robust evaluation of these hallucination-tackling methods over their baselines. It is our mistake that the formatting in the above comment is unclear and does not directly present the comparison between these methods and their respective baselines. The complete performances are as follows:
>
> Table 1: MCQ Accuracy of methods that tackle multimodal hallucination on the Hallucination Completion task of LongHalQA.
> | Accuracy | Description | Conversation | Average |
> |:---------------------|:--------------------------:|:--------------------------:|:--------------------------:|
> | LLaVA 1.5-7B   | 32.80	| 39.37 | 36.08        |
> | + POVID          | 36.25 |	40.86 | 38.55        |
> | + RLAIF           | 36.48	| 44.02 | 40.25        |
> |------------------------|------------------------|------------------------|------------------------|
> | LLaVA 1.5-13B    | 31.53  | 43.62 | 37.58        |
> | + LLaVA-RLHF   | 37.17 | 44.02 | 40.60        |
> |------------------------|------------------------|------------------------|------------------------|
> | Qwen-VL-Chat  | 33.14 | 40.00 | 36.57  |
> | + Silkie        | 37.17 | 40.63 | 38.90        |
> |------------------------|------------------------|------------------------|------------------------|
> | Muffin          | 17.15 | 26.46 | 21.80        |
> | +RLHF-V          | 31.99 |	31.02 |  31.50        |
>
> All the methods have robustly improved the baseline model on the hallucination completion task. Most methods primarily improve their performances on the description data, which aligns with their training from the image description task. Methods incorporating more diverse tasks and datasets, such as RLAIF, also achieve promising improvements for the conversation data. These comparisons demonstrate that LongHalQA can robustly evaluate the effectiveness of these methods in mitigating multimodal hallucinations.
>
> In contrast, the performance of hallucination-tackling methods under free-form completion, compared to LongHalQA, is indeed less robust. It is influenced by various factors, such as the model's instruction-following ability after re-tuning, changes in output length, and updates of the evaluator (e.g., GPT). Our MCQ completion task could largely avoid these issues and provide a more robust evaluation, supplementing the existing generative benchmarks with evaluation from external MLLMs.

---

> ### Author Response · Authors · 2024-11-24
>
> To better demonstrate the consistency between MCQ and generative evaluation, we include a free-generation image description task for comparison. This task is extensively trained across all MLLMs, reducing the influence of factors such as instruction-following ability. For each MLLM, we use the prompt "Describe the given image in detail." to generate image descriptions. We then feed the image, the reference description from LongHalQA, and object annotations from Objects365 to GPT, and ask GPT to score the MLLMs generated descriptions in terms of detail level and hallucination degree. Both scores range from 0 to 10, with higher scores indicating better performance(more detailed and less hallucination content). The results are shown in the table below.
>
> In Table 1, most MLLMs achieve similar rankings under MCQ(Desc. Acc.) and generation-based(Hall. Score) evaluations. The only differences in ranking appear with LLaVA-v1.6-7B and hallucination mitigation methods like RLAIL and RLHF-V. We argue that these differences primarily arise from differences in detail levels. Most hallucination mitigation methods rank much higher for hallucination scores than for detail scores in generative evaluations. MLLMs become more careful and expect to generate less detailed content to reduce hallucination risk.[1] In contrast, LLaVA-v1.6-7B outputs nearly twice as much content as other MLLMs, achieving the highest detail score but also producing more hallucinations in quantity. These differences highlight the importance of considering both detail and hallucination levels for a comprehensive evaluation of MLLMs' hallucination levels.
>
> On the contrary, our proposed MCQ evaluation directly queries MLLMs with the same challenging detailed content, eliminating the influence of varying detail levels of MLLMs in responding. The rankings from our MCQ evaluation also align more closely with the combined detail and hallucination scores from generative evaluations, especially for those hallucination mitigation methods. These results demonstrate the consistency between MCQ evaluation in LongHalQA and generative evaluation.
>
> Table 1. Comparison of MCQ evaluation and generative evaluation with the image description task. The last two columns are the performance and ranking of the MCQ hallucination completion task with image description data.
> |               |                 |    Free     |    Generated     |  Description  |                |                |     Hall     |   Completion   |
> |:--------------|:---------------:|:-----------:|:----------------:|:-------------:|:--------------:|:--------------:|:------------:|:--------------:|
> | **Methods**     | **Number of Words** | **Detail Rank** | **Detail Score** | **Hall Rank** |**Hall Score**| **Avg. Rank**  | **MCQ Rank** | **Desc.(MCQ)** |
> | MiniCPM-V2    |      135.3      |      2      |       6.45       |       3       |      6.66      |       2        |      2       |     44.07      |
> | Qwen2-VL-2B   |      107.6      |      3      |       6.34       |       2       |      6.78      |       1        |      1       |     47.18      |
> | LLaVA 1.6-7B  |      177.4      |      1      |       6.52       |       6       |      5.83      |       4        |      3       |     39.47      |
> | LLaVA 1.5-7B  |      104.8      |      8      |       5.46       |      10       |      5.38      |       9        |      9       |     32.80      |
> | + POVID       |      95.5       |     10      |       5.34       |       5       |      5.98      |       6        |      7       |     36.25      |
> | + RLAIL       |      104.1      |      7      |       5.52       |       1       |      6.80      |       4        |      6       |     36.48      |
> | LLaVA 1.5-13B |      102.9      |     11      |       5.33       |      11       |      5.25      |       11       |      11      |     31.53      |
> | + LLaVA-RLHF  |      116.5      |      6      |       5.54       |       7       |      5.78      |       6        |      4       |     37.17      |
> | Qwen-VL-Chat  |      97.3       |      5      |       5.70       |       9       |      5.55      |       8        |      8       |     33.14      |
> | + Silkie      |      99.2       |      4      |       6.23       |       4       |      6.34      |       3        |      4       |     37.17      |
> | Muffin        |      100.8      |      9      |       5.45       |      11       |      5.25      |       10       |      12      |     17.15      |
> | +RLHF-V       |      51.3       |     12      |       4.33       |       8       |      5.60      |       12       |     10       |    31.99     |
>
>
> [1] Yue Z, Zhang L, Jin Q. Less is more: Mitigating multimodal hallucination from an eos decision perspective[J]. arXiv preprint arXiv:2402.14545, 2024.

---

> ### Author Response · Authors · 2024-12-01
> **Looking forward to further discussion**
>
> We sincerely appreciate your insightful and valuable comments. Regarding the robustness of evaluating hallucination-mitigation methods, we would like to clarify that our LongHalQA robustly demonstrates their improvements in mitigating hallucinations, as shown in the tables in our latest two responses. The non-robust results are from our comparative experiments under the free-form text completion task (affected by model preference and instruction-following capabilities), not from our LongHalQA evaluation itself.  We have also supplemented comparisons of free-form generation and our MCQ approach on a more general image description task. Our MCQ and the free-generation setting demonstrate consistent improvements from these hallucination-mitigation methods.
>
> As the discussion phase is nearing its end, we kindly ask for a few minutes of your time to check our responses. If our responses have clarified your concerns, we hope you might consider raising your evaluation of our paper. We look forward to hearing from you about any further feedback or suggestions for improving our work.

---

### Official Review · Reviewer_pkPj · 2024-11-04

**Soundness:** 3
**Presentation:** 3
**Contribution:** 3
**Rating:** 8
**Confidence:** 3

**Summary:**

This paper proposes a long-context hallucination benchmark. This benchmark aims to solve two problems in the existing evaluation pipeline: it is too easy for discriminative tasks and too time-consuming for open-ended generative tasks. To achieve this, the authors propose the LongHalQA, which unifies discriminative and generative tasks as multi-choice problems. Also, they formulate the construction of LongHalQA as a pipeline to construct future hallucination benchmarks with long and complex questions and descriptions.

**Strengths:**

1. The paper is well-written and easy to follow.
2. The motivation is reasonable and practical. I think this benchmark will accelerate the development of MLLMs on hallucination.
3. The analysis of the experiment is relatively comprehensive.

**Weaknesses:**

1. A little small number of evaluated models.
2. No comparison between the performance of existing methods towards solving the hallucination of MLLMs. I'm interested in whether existing methods have improved on LongHalQA.
3. Lack of related work about the method about how to decrease the hallucination of MLLMs

**Questions:**

No

---

> ### Author Response · Authors · 2024-11-20
>
> We sincerely thanks for reviews' valuable feedback and the positive evaluation of our work. Below, we respond to each of the raised concerns.
>
> > 1. A little small number of evaluated models.
>
> Thank you for your suggestion. Due to the time constraint, we selected only a few widely adopted MLLMs for evaluation. Additionally, we skipped some MLLMs that do not support long-context scenarios due to limitations on context length, such as InstructionBLIP and MiniGPT4 series, which further limited our selection. We will continuously evaluate and report the results of other MLLMs on LongHalQA in the future.
>
> > 2. No comparison between the performance of existing methods towards solving the hallucination of MLLMs. I'm interested in whether existing methods have improved on LongHalQA.
>
> Thank you for your valuable suggestion. We evaluate several methods that aim at mitigating hallucinations in MLLMs as shown in the table below. We also show the performance of their baseline MLLMs for comparison. Most of these methods employ RLHF (Reinforcement Learning from Human Feedback) or DPO (Direct Preference Optimization) to refine MLLM output preferences and reduce hallucinations.
>
> Several points can be observed from our evaluations:
>
> (1) Most methods use image description tasks to construct preference data, which is well aligned with our observation that most methods significantly improve over the baseline model for image description in the completion task.
>
> (2) The preference optimization based on image description also effectively improves performances in discriminating hallucinations of object description, consistent with their gains observed on prior benchmarks such as POPE. However, these methods showed limited improvements for discrimination tasks under the long-context setting, which involves detailed image descriptions and multi-round conversation. When models are tasked to identify the reasons for hallucinations, their performances drop mostly on the hallucination discrimination task under multiple-choice question (MCQ) settings.
>
> (3) The preference optimization based on the image description task yields limited benefits for multimodal conversational capabilities. For example, CSR and POVID result in performance drops to around 20\% on discrimination tasks for the conversation. We found this is largely due to a decrease in instruction-following capabilities, where they fail to process long-context queries and correctly output the option letter. Furthermore, the improvements in hallucination completion tasks for conversation data are also limited compared to their baseline models.
>
> These findings highlight the current limitations of preference-based hallucination mitigation methods in addressing long-context scenarios, as well as the lack of diversity in tasks and data formats. We conjecture that improving the diversity and length of the constructed preference data has great potential to mitigate these issues.
>
> Table 1: Experiments on methods for mitigating hallucinations of MLLMs on LongHalQA. ''Obj.'', ''Des.'' and ''Con.'' are the data formats of object description, detailed image description, and multi-round conversations in LongHalQA. Columns 2-6 are the results of the hallucination discrimination task, and columns 7-8 are the results of the hallucination completion task.
>
> |                       |        |    Hallucination       |  Discrimination  | | | Hallucination | Completion |
> |:------------------|:------:|:------:|:------:|:------:|:------:|:------:|:------:|
> | **Methods**           |  **Obj.(Binary)**  |  **Des.(Binary)**  |  **Con.(Binary)**  |  **Des.(MCQ)**  |  **Con.(MCQ)**  | **Des.** | **Con.** |
> | LLaVA-1.5-7B   | 45.18  | 36.59  | 33.79  | 37.17  | 32.92  | 32.80  | 39.37  |
> | + CSR            | 44.52  | 36.59  | 31.30  | 36.66  | 20.16  | 32.86  | 36.54  |
> | + POVID          | 45.69  | 36.66  | 32.54  | 37.03  | 21.20  | 36.25  | 40.86  |
> | + RLAIF          | 60.29  | 36.88  | 35.79  | 35.28  | 30.17  | 36.48  | 44.02  |
> | LLaVA-1.5-13B | 52.70  | 36.95  | 35.85  | 45.99  | 41.21  | 31.53  | 43.62  |
> | + LLaVA-RLHF     | 48.83  | 36.66  | 36.91  | 41.11  | 37.78  | 37.17  | 44.02  |
> | Qwen-VL-Chat  | 58.69  | 36.66  | 34.29  | 37.97  | 36.10  | 33.14  | 40.00  |
> | + Silkie         | 62.19  | 36.95  | 35.22  | 37.54  | 35.29  | 37.17  | 40.63  |
> | Muffin-13B    | 46.50  | 52.99  | 67.46  | 38.85  | 30.92  | 17.15  | 26.46  |
> | + RLHF-V         | 60.29  | 38.19  | 40.90  | 38.12  | 25.62  | 31.99  | 31.02  |
>
> > 3. Lack of related work about the method about how to decrease the hallucination of MLLMs.
>
> Thank you for your suggestion. We will add a discussion on hallucination mitigation methods in the related work section.

---

### Official Review · Reviewer_vfgV · 2024-11-04

**Soundness:** 2
**Presentation:** 1
**Contribution:** 2
**Rating:** 3
**Confidence:** 4

**Summary:**

This paper proposes a new benchmark for evaluating hallucinations in multi-modal large language models (MLLMs).
The paper makes use of GPT4V to generate image-level and object-level descriptions and conversation data for a set of images from VisualGenome. These wider range of generated data enables the proposed benchmark, LongHalQA, to evaluate various types of potential hallucination which go beyond the typical object level analysis (e.g. Is there a cat in the image?). The proposed method suggests two types of evaluation: (1) Hallucination Discrimination - the model must answer a MCQ about generated data (potentially containing hallucinations), to determine if the generated data contains hallucinations based on the image and the cause of the hallucination if present; (2) Hallucination Completion - the model must answer a MCQ, correctly selecting the answer which truthfully completes a partial conversation or description.
The authors conduct experiments on a range of open-source MLLMs and the closed-source GPT4o. They show that CoT prompting often has little or negative effect on results on LongHalQA. Finally they conduct a study in which hallucinations in free-form generations from their questions yield similar results to using their MCQ formulation.

**Strengths:**

This paper correctly identifies that many prior hallucination works focus on the narrow topic of object existence at an image level. To overcome they create questions which expand the evaluation to object level descriptions, object locations, attributes etc.

Their experimental results are numerous and allow the reader see the advantages/disadvantages of each model in the different types of question in LongHalQA (Table 2-5).

The authors make comparisons of their MCQ method to a free-form generation method in Section 6 and demonstrate the advantages of using MCQ over a free-form method in terms of efficiency of evaluation.

**Weaknesses:**

I have two main weaknesses with this paper, unfortunately both of which I consider pretty major.

### 1. The logic behind the creation of the benchmark itself.

As detailed in Section 4, all of the LongHalQA data comes from generations with GPT4V, this includes the descriptions, conversations etc. These generations are then analysed/modified with a number of checks. Furthermore, the question options themselves are generated with GPT4V. Therefore when evaluating a model X using LongHalQA, you are conditioning all reasoning/grounding/recognition of model X on the range of hallucinations GPT4V might make. This leaves a large range of potential hallucinations that are specific to model X which are left to be analysed, which may only be obtained by generating descriptions/conversations using model X rather than GPT4V. Taking Figure 1, GPT4V and the method used in Section 4 have created a hallucination regarding the number of people seated in the carriage. Now this is a hallucination of GPT4V + Section 4, _not_ of model X. Model X may have hallucinated the species of animal, the colour of the carriage etc, all of which is left potentially undiscovered because the hallucinations model X is asked to evaluate in LongHalQA are not its own, I therefore find the logic of this benchmark slightly confused. The free-form generations of methods like that of Kaul et al. and Jiang et al. referenced in the paper need the model being evaluated e.g. Model X to _actually generate_ its own text and therefore its own potential hallucinations.

### 2. Lack of details and clarity.

The crucial step in this work is the generation of the data for LongHalQA, detailed in Section 4. I find this section to be extremely thin on details and lack clarity.
1. L291 "...then analyze and filter them based on dataset annotations and GroundingDINO...", no information is given on how this process is done.
2. L297, "as illustrated in Appendix B." Appendix B contains a list of definitions of hallucinations used in this work.
3. L303, "Second, names of object present in the data are extracted, and certain image understanding tools such as GroundingDINO...", there are no details on how objects present in the data are extracted, which data? VG annotations or names in the GPT4V generated data or both? Which image understanding tools other than GroundingDINO are used?
3. L314-319, GPT4V is being used to generated hallucination explanation pairs, but there is no indication that manual checking is used here despite the authors accepting that GPT4V suffers from "sever hallucinations" (L298), the logic here seems confused on the ability of GPT4V to create such specific data which only contains one error which is also useful for evaluation.
4. L320-L346, same arguments as above with the ability of GPT4V to this accurately.
5. L344 "except the hallucination checking that involves optional human verification" does this mean human verification is used or not? What is the effect of using human verification in the data vs not?

Additionally as a more general point, the prompt templates used in Appendix C are extremely hard to follow without any examples, e.g. in Figure 6 what is "Possible Content"? The main text asks the reader to refer to Appendix C (L465) for details and then appears to simply paste the prompts used with no explanation of what goes where.

**Questions:**

### 1. Logic Behind the Benchmark Creation
1. Since all LongHalQA data is generated by GPT-4V, isn’t Model X limited to analysing GPT-4V’s specific hallucinations rather than its own?
2. Could Model X be missing its unique hallucinations because it doesn’t generate its own descriptions or conversations in LongHalQA?
3. Wouldn’t a model evaluation approach where Model X generates its own text reveal more relevant hallucinations, as done in Kaul et al. and Jiang et al.?

### 2. Lack of Details and Clarity
1. How are dataset annotations and GroundingDINO used to filter the LongHalQA data? Can details on this process be provided?
2. How are objects identified in the data? Are these from VG annotations, GPT-4V data, or both?
3. Other than GroundingDINO, which image understanding tools are used, and how?
4. If GPT-4V produces hallucination explanation pairs, is there manual verification, especially given its acknowledged hallucination issues (L298)?
5. In what cases is human verification used for hallucination checking, and how does it impact the dataset?

### Additional Clarity
1. Can examples be given to the prompt templates in Appendix C to clarify instructions like "Possible Content" etc. (Figure 6)?

---

> ### Author Response · Authors · 2024-11-20
>
> We sincerely thank you for your valuable comments, which are exceedingly helpful in improving our paper. Our responses to each comment are as follows.
>
> ### **Reply to Logic Behind the Benchmark Creation**
>
> >1. Since all LongHalQA data is generated by GPT-4V, isn’t Model X limited to analysing GPT-4V’s specific hallucinations rather than its own?
> >
> >2. Could Model X be missing its unique hallucinations because it doesn’t generate its own descriptions or conversations in LongHalQA?
>
> As one of the most advanced MLLMs, the GPT-4V-generated hallucinations can well represent visually misleading content in images. Therefore, our motivation is to gauge the hallucination levels of MLLMs by evaluating their capability to directly describe such challenging content. As shown in Table 5, MLLMs indeed struggle to distinguish between hallucinations and correct options when completing these challenging contents. The poor accuracy (mostly below 50\%) indicates that these hallucinations are not specific to GPT-4V but rather a common problem faced by most MLLMs. We agree that incorporating more models would provide a more comprehensive view of MLLM hallucination. We plan to employ other powerful models beyond GPT, incorporating more challenging hallucination content generated by various MLLMs to further enhance the diversity of the LongHalQA.
>
> We would highlight that the completion task in LongHalQA covers 12 different types of hallucinations and includes 2,139 long-context samples, with completion options featuring three different possible hallucinations. This diverse set of queries is able to capture most types of potential hallucinations the tested model X might generate.
>
> >3. Wouldn’t a model evaluation approach where Model X generates its own text reveal more relevant hallucinations, as done in Kaul et al.[1] and Jiang et al.[2]?
>
> We believe that our LongHalQA and existing generative evaluation benchmarks each have complementary strengths.
>
> Existing generative hallucination benchmarks, such as those by Kaul et al.[1] and Jiang et al.[2], provide a straightforward evaluation of MLLMs generated content. However, these benchmarks have two clear constraints: 1) reliance on external MLLM evaluators; 2) limited scope of evaluated hallucinations. For example, Kaul et al.[1] use the FLAN-T5 model and can only detect simple object-existing hallucinations. Jiang et al.[2] build 2M data for fine-tuning the LLaVA-1.5-13B model as an evaluator for hallucinations related to objects, relationships, attributes, and events. When other types of hallucinations or data from other domains are introduced, the LLM evaluator must be re-trained, restricting the benchmark's flexibility and applicability. Additionally, benchmarks based on MLLM evaluators may suffer from the evaluators' randomness, as well as low efficiency for both generating and evaluating long descriptions.
>
> As a comparison, by presenting potential hallucination content as completion options, LongHalQA directly evaluates the hallucination levels of MLLMs on these challenging contents, leading to a more challenging assessment while covering more diverse types of hallucinations. Additionally, the MCQ format simplifies the testing process, making it easier to obtain definitive evaluation results. Further, LongHalQA provides detailed assessments of complex hallucinations in long-context scenarios, excelling in efficiency, scalability, and diversity. We believe LongHalQA can complement existing benchmarks in evaluating long, complex hallucinations of MLLMs.
>
>
> [1] Kaul P, Li Z, Yang H, et al. THRONE: An object-based hallucination benchmark for the free-form generations of large vision-language models[C]//Proceedings of the IEEE/CVF Conference on Computer Vision and Pattern Recognition. 2024: 27228-27238.
> [2] Jiang C, Jia H, Dong M, et al. Hal-eval: A universal and fine-grained hallucination evaluation framework for large vision language models[C]//Proceedings of the 32nd ACM International Conference on Multimedia. 2024: 525-534.

---

> > ### Author Response · Authors · 2024-11-25
> >
> > > **Consistency between the hallucination completion task of LongHalQA and typical generative evaluation.**
> >
> > In Table 8 of the paper, we compare free-form generation and MCQ evaluations, showing that the rankings are largely consistent across both settings. To further demonstrate this consistency, we conduct additional experiments using the typical image description task to assess MLLM hallucination levels while also introducing more methods, such as those designed to mitigate multimodal hallucinations. For each MLLM, we use the prompt "Describe the given image in detail." to generate image descriptions. We then feed the image, the reference description from LongHalQA, and object annotations from Objects365 to GPT, and ask GPT to score the MLLMs generated descriptions in terms of detail level and hallucination degree. Both scores range from 0 to 10, with higher scores indicating better performance(more detailed and less hallucination content). The results are shown in the table below.
> >
> > The table below shows that LongHalQA achieves rankings consistent with generative evaluations that consider both hallucination levels and detailedness. This indicates that directly using visually challenging and hallucination-prone content to query MLLMs can yield results similar to directly evaluating their generated content. Such challenging content is likely to induce hallucinations in most MLLMs, making it effective in being used to evaluate the hallucination level of MLLMs. In the future, we plan to collect more hallucinated content from advanced MLLMs to expand LongHalQA further.
> >
> > Table 1. Comparison of MCQ evaluation and generative evaluation with the image description task. The last two columns are the performance and ranking of the MCQ hallucination completion task with image description data.
> > |               |                 |    Free     |    Generated     |  Description  |                |                |     Hall     |   Completion   |
> > |:--------------|:---------------:|:-----------:|:----------------:|:-------------:|:--------------:|:--------------:|:------------:|:--------------:|
> > | **Methods**     | **Number of Words** | **Detail Rank** | **Detail Score** | **Hall Rank** |**Hall Score**| **Avg. Rank**  | **MCQ Rank** | **Desc.(MCQ)** |
> > | MiniCPM-V2    |      135.3      |      2      |       6.45       |       3       |      6.66      |       2        |      2       |     44.07      |
> > | Qwen2-VL-2B   |      107.6      |      3      |       6.34       |       2       |      6.78      |       1        |      1       |     47.18      |
> > | LLaVA 1.6-7B  |      177.4      |      1      |       6.52       |       6       |      5.83      |       4        |      3       |     39.47      |
> > | LLaVA 1.5-7B  |      104.8      |      8      |       5.46       |      10       |      5.38      |       9        |      9       |     32.80      |
> > | + POVID       |      95.5       |     10      |       5.34       |       5       |      5.98      |       6        |      7       |     36.25      |
> > | + RLAIL       |      104.1      |      7      |       5.52       |       1       |      6.80      |       4        |      6       |     36.48      |
> > | LLaVA 1.5-13B |      102.9      |     11      |       5.33       |      11       |      5.25      |       11       |      11      |     31.53      |
> > | + LLaVA-RLHF  |      116.5      |      6      |       5.54       |       7       |      5.78      |       6        |      4       |     37.17      |
> > | Qwen-VL-Chat  |      97.3       |      5      |       5.70       |       9       |      5.55      |       8        |      8       |     33.14      |
> > | + Silkie      |      99.2       |      4      |       6.23       |       4       |      6.34      |       3        |      4       |     37.17      |
> > | Muffin        |      100.8      |      9      |       5.45       |      11       |      5.25      |       10       |      12      |     17.15      |
> > | +RLHF-V       |      51.3       |     12      |       4.33       |       8       |      5.60      |       12       |     10       |    31.99     |

---

> ### Author Response · Authors · 2024-11-20
>
> ### **Reply to Lack of Details and Clarity**
>
> We sincerely appreciate your pointing out these issues. We provide further elaborations below and will revise the paper to clarify these details.
>
> > 1. How are dataset annotations and GroundingDINO used to filter the LongHalQA data? Can details on this process be provided?
>
> We leverage GroundingDINO to remove those inaccurate annotated boxes labeled as "crowd" and remove images that lack sufficient complexity or richness of content~(such as containing less than five objects).
>
> > 2. How are objects identified in the data? Are these from VG annotations, GPT-4V data, or both?
>
> Here, we employ the GPT to extract the object phrases from its generated data, such as "horse-drawn carriage", "brown horse", and "three passengers in the carriage" as illustrated in Figure 1. These object phrases are fed into GroundingDINO to detect relative bounding boxes. We upload the detection results to employ GPT to check the object-related hallucinations in the generated data.
>
> > 3. Other than GroundingDINO, which image understanding tools are used, and how?
>
> We mainly adopt GroundingDINO and GPT itself in checking the GPT-generated data. We used GPT to conduct multiple rounds of sentence-level verification using different prompts. These prompts include grounding results from GroundingDINO and potential hallucination types, as outlined in Table 1. Finally, we employ GPT to summarize the verification results and provide a report indicating whether each sentence in the generated data contained hallucinations and the corresponding analysis.
>
> > 4. (1) L314-319, GPT4V is being used to generated hallucination explanation pairs, but there is no indication that manual checking is used here despite the authors accepting that GPT4V suffers from "sever hallucinations" (L298), the logic here seems confused on the ability of GPT4V to create such specific data which only contains one error which is also useful for evaluation. (2) L320-L346, same arguments as above with the ability of GPT4V to this accurately.
>
> We would like to clarify that the severe hallucinations mentioned in L298 specifically refer to multimodal hallucinations produced by GPT when describing image content. However, GPT's text processing capability remains reliable. In L314-346, we mainly leveraged GPT's text-processing abilities to construct hallucination explanation pairs, as well as questions and options. GPT only needs to modify text based on sentence-level analysis results and format it according to LongHalQA's requirements, which is rarely affected by multimodal hallucination issues.
>
> > 5. (1) If GPT-4V produces hallucination explanation pairs, is there manual verification, especially given its acknowledged hallucination issues (L298)? (2) In what cases is human verification used for hallucination checking, and how does it impact the dataset? (3) L344 "except the hallucination checking that involves optional human verification" does this mean human verification is used or not? What is the effect of using human verification in the data vs not?
>
> We deploy human verification for the sentence-level hallucination analysis as summarized by GPT. In the hallucination check process, we leverage GPT to label each sentence in the generated data as either "Match" or "Do not Match," along with corresponding hallucination explanations, as illustrated in Figure 7. Human evaluators then review the data and determine whether to accept the analysis. The verification process for most data is quite efficient, typically taking about one or two minutes. If the analysis is not accepted, the human evaluator needs to revise the label or the analysis. This step is crucial to ensure the correctness of our data. Though simpler object or attribute hallucinations can be mostly identified by previous multi-round checks and the help of the object detector, more complex hallucinations, such as those shown in Figure 2, require the human evaluator for verification. These human-verified complex hallucinations also enrich the diversity of hallucinations covered in LongHalQA and increase the overall difficulty of the benchmark.
>
> > 6. Can examples be given to the prompt templates in Appendix C to clarify instructions like "Possible Content" etc. (Figure 6)?
>
> In Figure 6, we first provide a prompt template, followed by specific examples of prompts for different data formats (e.g., object descriptions, image descriptions, multi-turn conversations). The "Possible Content" denotes "object types, colors, states, actions, number of objects, precise object locations, texts or OCR results, relationships or relative positions between objects, etc.". We will update Figure 6-9, as well as their captions, to make the presentation clearer.

---

> ### Author Response · Authors · 2024-12-01
> **Looking forward to further discussion**
>
> We sincerely appreciate your insightful and valuable comments. We have carefully addressed the main concerns in detail through experiments and explanations. We kindly ask for a few minutes of your time to check our responses.  If our responses have clarified your concerns, we hope you might consider raising your evaluation of our paper. As the discussion phase is about to close, we look forward to hearing from you about any further feedback or suggestions for improving our work. We will be very happy to clarify any further concerns (if any).

---

> ### Comment · Reviewer_vfgV · 2024-12-01
> **Issues remain**
>
> While I appreciate the responses to the points I have raised, I do believe that many of the questions remain:
>
> ### Logic of the benchmark
> I still maintain that using GPT4V to _generate_ hallucinations does not make it possible to evaluate a given model's hallucinations.
> The authors propose to ask an MLLM of interest to determine hallucinations amongst a set of answers generated by another model (GPT4V). Taking Figure 1 as an example: the description provided contains numerous concepts that an MLLM of interest _given the chance_ may hallucinate about: (a) species of the animal drawing the carriage; (b) colour of the animal; (c) colour of the carriage; (d) colour of the wheels; (e) clothes of the man (I suspect GPT4V might've incorrect said 'vest' instead of 'shirt') etc. etc.
> None of these potential hallucinations of a model are being analysed in this work because the model has never been asked to or given the "freedom" to hallucinate.
>
> Moreover, I agree with the point raised by Reviewer Xq4o that one of the two tasks "Hallucination Completion" is still not really generative and really asks the model to discriminate between options provided by GPT4V. This counts against the authors' claim that they are doing generative evaluation of an MLLM of interest.
>
> Finally, you rightly argue that previous methods require LLM evaluators and additional inference time to generate MLLM hallucinations, but I would note that this work requires human effort to manually verify that only one hallucination has been made and that the hallucinations are correct. The former is much more scalable than the latter as LLM evaluators are becoming more powerful for the same compute budget and the cost to generate a token from a given LLM is likely to decrease with time.
>
> While I agree with the authors that the _evaluation_ process of the method is relatively simple because it is in effect a complex MCQ benchmark. The _creation_ process requires human effort for each potential question and I am not convinced the method is actually measures what a hallucination benchmark should. Particularly with regards to the claim of "generative evaluation", this claim is fundamentally wrong. On the above basis I am reluctant to change my score, despite the clear effort from the authors.
>
> ### Details
> Thank you to the authors for providing clarifications. I believe that all of that which has been provided should be in the main body of the paper (and more), the details of how benchmarks are created and used is important for the readers alongside clear code. I do not appear to see any paper revisions which incorporates these additional details and I personally think this work could do with more time to be structured such that these important details are included.
>
> I thank the authors for their efforts, but I do not believe the fundamental concerns I raised have been addressed.

---

### Meta-Review · Area_Chair_Vk4J · 2024-12-22

**Metareview:**

The paper introduces LongHalQA, a new benchmark for evaluating hallucinations in multi-modal large language models (MLLMs). Hallucinations occur when a model generates text that misrepresents the input image. The benchmark tries to addresses the key limitations in existing evaluation methods: the simplicity of discriminative tasks and the inefficiency of open-ended generative tasks. LongHalQA consists of 6,000  complex hallucinatory texts that mimic real-world scenarios,  formalized in two tasks types: hallucination discrimination and hallucination completion,  which combine both discriminative and generative evaluations into a single multiple-choice format. The authors conduct experiments on a range of open-source MLLMs and the closed-source GPT4o.

Reviewers agree LongHalQA is investigating an important direction of the field, and the motivation is reasonable and practical. Concerting the problem to a single multi-choice format climate the need of LLM, which makes the benchmark more stable and reliable.

However, reviewers are also concerned about the generation process, such as using GPT4V to generate hallucinations does not make it possible to evaluate a given model's hallucinations, the lack of verification of the benchmark.  In addition, two of the reviewers also concerned about the claim of considering the multi-choice question as a generative evaluation but not a discriminative evaluation.

**Additional Comments On Reviewer Discussion:**

The authors are engaged in the discussion period and provided more details about the generation process. However, the major concerns remains and additional work is needed to reach the ICLR standard.

---

### Decision · Program_Chairs · 2025-01-22

Reject